# LRRC59 cooperates with nuclear transporters to restrain the nuclear envelope repair machinery and safeguard genome integrity

Romy Timmer [1], Aurélie Bellanger [1], Sarah Peeters [1,2], Hera Kim [1], Laura Rodriguez de la Ballina [3,4], Sissel Eikvar[1], Annemijn J. Arns[1], Esmée Oortgijs [1], Nikolina Sekulić[1], Winnok H. De Vos [2,5,6] & Coen Campsteijn [1] ✉

Nuclear envelope (NE) rupture is a hallmark of cancer cells, and persistent NE damage drives genome instability and inflammation. NE repair relies on activation of the endosomal sorting complex required for transport (ESCRT)-III repair machinery by the LEMD2-CHMP7 compartmentalization sensor, but little is known beyond these core factors. Here, we use convergent proximity proteomics to inventorise proteins mobilized to the NE upon assembly of LEMD2-CHMP7 and activation of ESCRT-III. Within this NE repairome, we identify LRRC59 as a critical regulator of LEMD2 accumulation at NE ruptures. We find that LRRC59, together with the nuclear transporters KPNB1 and XPO1, restricts the assembly of LEMD2-CHMP7 complexes to the site of rupture. Disruption of this regulatory axis escalates LEMD2-CHMP7 spreading across the NE, driving torsional DNA damage in ruptured nuclei and micronuclei. Thus, our work identifies a central regulatory layer of NE repair centered on LRRC59 and KPNB1. We propose that altered LRRC59 levels and deregulated nuclear transport coordinately compromise NE repair, driving genome instability and cancer development.

The nuclear envelope (NE) serves as a vital protective barrier for the genome and comprises a double phospholipid bilayer, highly selective nuclear pore complexes, and the nuclear lamina. Work over the last decade has shown that the NE undergoes transient ruptures during interphase[1-5], in models of viral infection[6], aggressive cancers in vitro[7] and in vivo[8,9], genetic nuclear fragility[10], or upon exposure to biomechanical forces[2,11-13]. These NE ruptures cause nucleocytoplasmic mixing, compromise nuclear function and genome integrity, and

activate pro-inflammatory signaling cascades[1-5] – all major challenges to cell fitness.

To prevent or minimize lasting damage, NE rupture sets into motion a series of molecular mechanisms that detect NE lesions and ensure acute and coordinated restoration of the nuclear barrier within minutes[1,14]. Following NE rupture, the barrier-to-autointegration factor (BAF) rapidly accumulates at the exposed chromatin and recruits key repair factors, including the inner nuclear membrane (INM)

[1]Department of Molecular Medicine, Institute of Basic Medical Sciences, University of Oslo, Oslo, Norway. [2]Laboratory of Cell Biology and Histology, Department of Veterinary Sciences, University of Antwerp, Antwerp, Belgium. [3]Centre for Cancer Cell Reprogramming, Faculty of Medicine, University of Oslo, Oslo, Norway. [4]Department of Molecular Cell Biology, Institute for Cancer Research, Oslo University Hospital, Oslo, Norway. [5]Antwerp Centre for Advanced Microscopy, University of Antwerp, Antwerp, Belgium. [6]µNeuro Research Centre of Excellence, University of Antwerp, Antwerp, Belgium. ✉e-mail: Coen.campsteijn@medisin.uio.no

transmembrane protein LEMD2[11,15]. Concurrently, loss of NE integrity is detected by a compartmentalization sensor composed of LEMD2 and the ESCRT-III protein CHMP7. Under normal conditions, CHMP7 localization is restricted to the ER by virtue of its N-terminal membrane-binding domain[16] and active Exportin 1 (XPO1)-mediated nuclear export[17,18]. However, loss of NE integrity results in the local dissipation of the RAN-gradient[19], disrupting XPO1-mediated export of CHMP7 and enabling formation of LEMD2-CHMP7 complexes at the site of NE rupture[17,18,20–22]. LEMD2-CHMP7 association[23] triggers the assembly of the ESCRT-III machinery that drives membrane fission and restoration of NE integrity through a series of tightly controlled steps[14,24,25].

Micronucleation, the formation of micronuclei (MN) from mis-segregating chromosomes, is frequently observed in cancers and prevalence of MN is associated with poor prognosis[26–28]. Micronuclei are prone to NE rupture and structural collapse[29], resulting in chromothripsis - extensive chromosomal rearrangements on individual chromosomes[30–33], and cGAS-driven pro-inflammatory signaling[34–37]. These phenomena are thought to be key factors in driving cancer development and metastasis, as well as resistance to treatments[26–28]. Unlike primary nuclei (PN), MN do not commonly recover from NE rupture, even though they recruit the ESCRT-III machinery after rupture[18]. This lack of repair is thought to stem from unbalanced LEMD2-CHMP7 interaction and unrestrained ESCRT-III activity, causing extensive DNA damage and micronuclear collapse[18]. In support of this model, elevated levels of reactive oxygen species (ROS) from nearby mitochondria inhibit CHMP7 export and drive its interaction with LEMD2, resulting in micronuclear membrane deformation and collapse[38,39]. Furthermore, ROS recruits the key autophagy factor p62 to ruptured MN, which degrades essential ESCRT-III members and thereby further impairs MN repair[39]. Why this imbalance is prevalent at MN but not PN remains poorly understood.

Despite an emerging framework, we still lack a comprehensive understanding of the signaling cascades and their spatiotemporal orchestration during NE rupture and repair. Unbiased identification and characterization of regulators and associated processes at these ruptures is complicated by the transient and sporadic nature of these events. Here we combine APEX2-based proximity proteomics of NE repair proteins LEMD2 and CHMP4B with inducible CHMP7 alleles to compile a comprehensive proteome associated with these NE repair complexes. Within this NE repair, we show that the leucine-rich repeat-containing 59 (LRRC59) constitutes a key layer of regulation of the LEMD2-CHMP7 compartmentalization sensor, together with the importin KPNB1, and in synergy with XPO1. Deregulation of this regulatory axis results in unrestrained spread of LEMD2 from the site of rupture, which triggers widespread DNA torsional stress and compromised cell fitness. We further show that LRRC59 controls MN repair, and its dysfunction results in the accumulation of DNA damage in rupturing MN. Together, our data uncover an LRRC59-centered regulatory axis of NE repair and provide a highly specific inventory of NE rupture responders.

## Results

### Establishment of a cell system to identify regulators of NE repair

While work from various labs[11,12,15,18,21–23,40–42] has established a molecular framework centered around ESCRT-III function that responds to resolve compromised NE integrity, very little is known beyond this core machinery. To establish a controllable system that enables comprehensive identification of proteins associated with NE repair, we engineered retinal pigment epithelial (RPE1) cells harboring one of three doxycycline (DOX)-inducible CHMP7 alleles: wildtype CHMP7[WT], nuclear localization signal (NLS)-fused CHMP7[NLS], or CHMP7[NES*] mutated in its nuclear export signal (NES)[18]. Induction of CHMP7 expression in these cells induces assembly of the LEMD2-CHMP7 nuclear compartmentalization sensor and triggers subsequent recruitment of the ESCRT-III machinery[18], recapitulating key elements of NE repair. In this background, we stably introduced fluorescent (mCitrine) engineered

ascorbate peroxidase APEX2-fusions of either the inner nuclear membrane (INM) protein LEMD2 or the soluble ESCRT-III subunit CHMP4B. Together, these systems enabled us to perform APEX2-based proximity proteomics to map the molecular neighborhoods of LEMD2 and CHMP4B, and identify factors associated with LEMD2-CHMP7-CHMP4B foci following induction of the *CHMP7* transgene (Fig. 1a).

To validate the system, we monitored the localization of constitutively expressed APEX2-mCitrine fusions of LEMD2 and CHMP4B upon DOX-induction of each of the three CHMP7 alleles. Consistent with our previous findings[18], expression of CHMP7[WT] caused a gradual relocation of LEMD2 and CHMP4B fusions to ectopic foci along the endoplasmic reticulum (ER) (Fig. 1b). In contrast, CHMP7[NLS] and CHMP7[NES*] triggered formation of NE-associated LEMD2-CHMP7-CHMP4B foci within hours[18] (Fig. 1b, c). Importantly, proximity biotinylation (Methods) in these models showed strong labeling of proteins (Supplementary Fig. 1A, B) proximal to LEMD2 and CHMP4B, most notably in foci upon CHMP7 induction (Fig. 1b and c; Strep-AF).

With this system in place, we performed two sets of proteomics experiments. Firstly, we performed proximity proteomics of APEX2-LEMD2 or APEX2-CHMP4B fusions in the absence or presence of induced CHMP7[WT] (12 h) or CHMP7[NLS] (6 h; Fig. 1b) to generate spatially resolved protein inventories of ER- vs NE-localized foci, respectively. Importantly, label-free quantitative (LFQ) LC-MS/MS analysis of these pulldowns showed a strong and selective enrichment of CHMP7 specifically after DOX induction and when compared to parental cell lines lacking APEX2-fusions (Fig. 1d). Furthermore, CHMP7 induction was required for the enrichment of endogenous CHMP4B and LEMD2 in pulldowns from LEMD2-APEX2 and CHMP4B-APEX2 cells, respectively (Fig. 1e and 1f, respectively). Taken together, these results highlight the central role of CHMP7 in the formation of ternary LEMD2-CHMP7-CHMP4B complexes and provided further critical validation for our experimental setup. Importantly, using CHMP7 alleles that target the ER and INM allowed us to subclassify proteomics hits based on subcellular localization (CHMP7[WT] vs. CHMP7[NLS], respectively).

Secondly, we generated proteomic snapshots at key timepoints after *CHMP7* induction. We focused on CHMP7[NES*] to permit optimal temporal separation of cofactor recruitment in relation to loss of NE integrity[18]. Previously, we have shown that expression of the CHMP7[NLS] or CHMP7[NES*] alleles compromises NE integrity, likely by imposing torsional stress locally on the NE[18]. To relate the induction of CHMP7[NES*] to the onset of NE ruptures, we used live-cell imaging of an NES-mRuby3 nuclear integrity marker (Fig. 1g and h). Based on this, we generated proteomics snapshots at 0 h, 3 h, 6 h (low NE rupture fraction), and 9 h (high NE rupture fraction) after CHMP7[NES*] induction (Fig. 1c, g and h). As above, we observed a strong and time-dependent enrichment of CHMP7 in pulldowns from APEX2-LEMD2 and APEX2-CHMP4B cells (Fig. 1I). Furthermore, the interaction of CHMP4B and LEMD2 again critically relied on CHMP7[NES*] expression (Fig. 1j and k). As such, our combined proteomics pipelines allowed us to map protein enrichment at NE lesion sites with spatial and temporal resolution.

### Proximity proteomics reveals a highly specific NE repairome

Extending beyond the core regulators of NE repair - CHMP7, LEMD2, and CHMP4B -, we identified over 3000 proteins in our proteomics dataset of which over 2000 were significantly enriched compared to parental cell lines lacking APEX2-fusions (Fig. 2a; Methods). As LEMD2 and CHMP4B do not interact in the absence of CHMP7 (Fig. 1) and normally display distinct localization patterns (Fig. 1), we expected their proteomes to overlap primarily following CHMP7 induction. Indeed, principal component analysis (PCA) of the CHMP7[NES*] time course showed a progressive convergence of LEMD2 and CHMP4B proteomes with prominent clustering from 6 h CHMP7[NES*] induction onwards (Fig. 2b).

Taking advantage of the spatial component of our experimental setup, we asked whether specific proteins would only mobilize to the

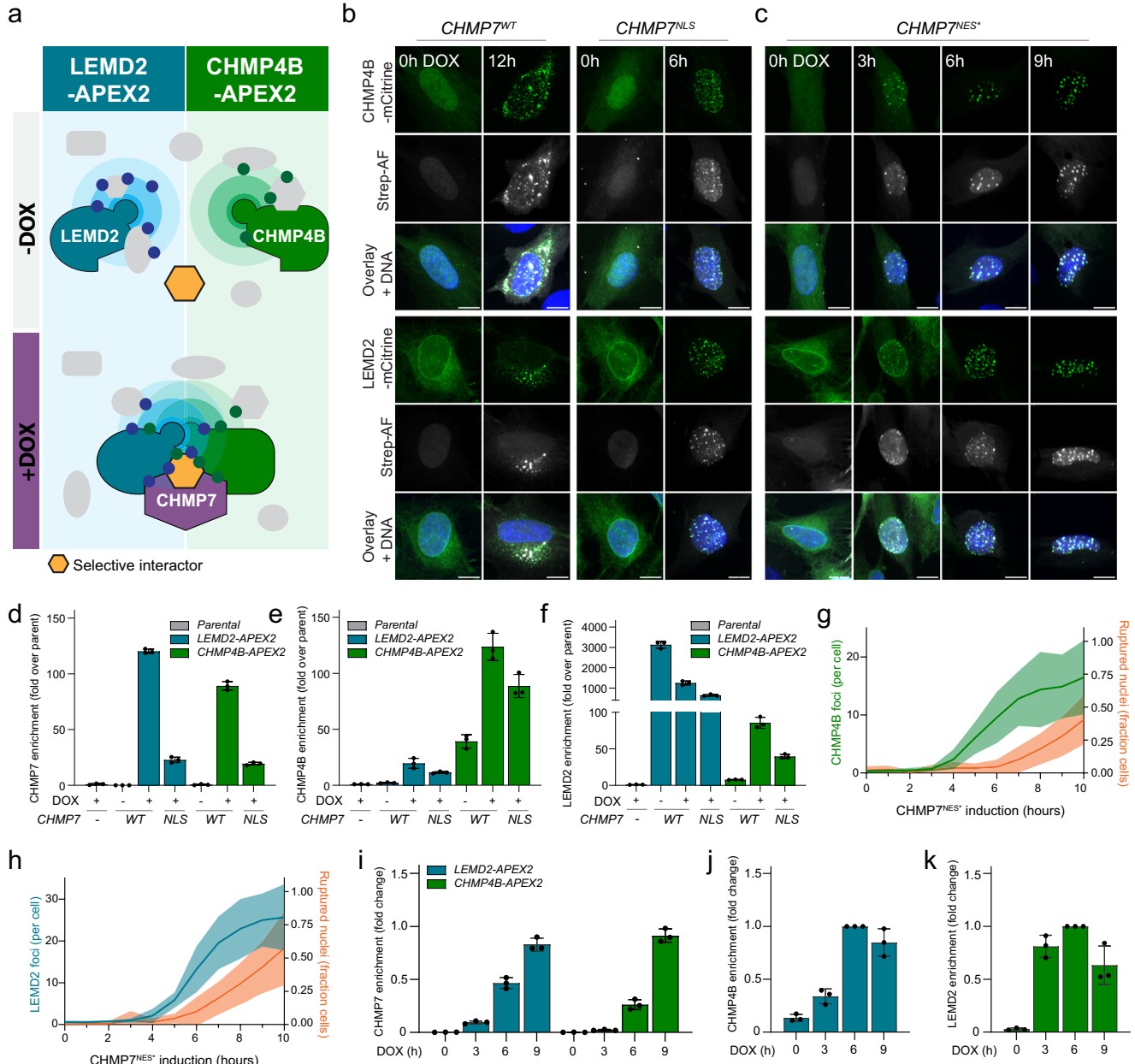

**Fig. 1 | Nuclear rupture proximity labeling system and spatiotemporal mapping of core repair factors. a** Schematic showing proximity labeling system to identify NE rupture-associated proteins. RPE1 cells lines stably expressing LEMD2-APEX2-mCitrine or CHMP4B-APEX2-mCitrine as well as DOX-inducible wildtype, NLS-fused, or NES-mutated CHMP7 alleles. This system allows for biotinylation of proximal proteins in the absence or presence of CHMP7 expression.
**b** Representative confocal images showing localization of LEMD2-APEX2-mCitrine or CHMP4B-APEX2-mCitrine (as indicated) in addition to biotinylated proteins (Strep-AF) following DOX treatment for the indicated timepoints to induce either CHMP7[WT] (left panels) or CHMP7[NLS] (right panels). DNA was stained with Hoechst. Scale bars, 10 μm; *N* = 3. **c** As panel **b**, but now after DOX treatment to induce the CHMP7[NES*] allele. Scale bars, 10 μm; *N* = 3. **d**–**f** Proteomics enrichment of CHMP7 (**d**), CHMP4B (**e**), or LEMD2 (**f**) following induction of CHMP7[WT] or CHMP7[NLS] in cells expressing LEMD2-APEX2 (blue) or CHMP4B-APEX2 (green). Cells were treated with

DOX to induce CHMP7[WT] or CHMP7[NLS] alleles and labeled with biotin-phenol. Error bars: mean ± SD. Each dot represents the mean per technical replicate; *N* = 3.
**g**, **h** CHMP7[NES*] overexpression results in formation of CHMP4B (**g**) or LEMD2 (**h**) foci followed by NE rupture. RPE1 LEMD2-APEX2-mCitrine or CHMP4B-APEX2-mCitrine cells were treated with DOX to induce CHMP7[NES*] expression. Number of foci (green or blue, left *y*-axis) and fraction of ruptured nuclei (orange, right *y*-axis) was scored every hour up to 10 h. Ruptured nuclei were identified by nuclear mRuby3-NES influx. Data represent means (lines) ± SD (bands) from 10 fields, *N* = 2. **i**–**k** Proteomics enrichment of CHMP7 (**i**), CHMP4B (**j**), or LEMD2 (**k**) following CHMP7[NES*] induction in cells expressing LEMD2-APEX2 (blue) or CHMP4B-APEX2 (green). Cells were DOX-treated to induce CHMP7[NES*] expression (0 h, 3 h, 6 h, 9 h) and labeled with biotin-phenol. Error bars: mean ± SD. Each dot represents the mean per technical replicate; *N* = 3. Source data are provided as a Source Data file.

NE upon assembly of the LEMD2-CHMP7 compartmentalization sensor and its downstream CHMP4B effector. To do this, we compared enriched proteins in the NE-resident CHMP7[NLS] vs ER-targeted CHMP7[WT] proteomes. This identified only 3 proteins that were specifically enriched ( > 5-fold relative enrichment) at nuclear LEMD2-CHMP7-

CHMP4B foci, namely Topoisomerase 2B (Top2B), Vaccinia-related kinase 1 (VRK1), and ALYREF (Fig. 2c). Top2B is recruited to chromatin to resolve DNA torsional stress[43], and we have previously shown that Top2B is selectively recruited to sites of excessive LEMD2-CHMP7-CHMP4B accumulation at the NE[18]. VRK1 has been implicated in the

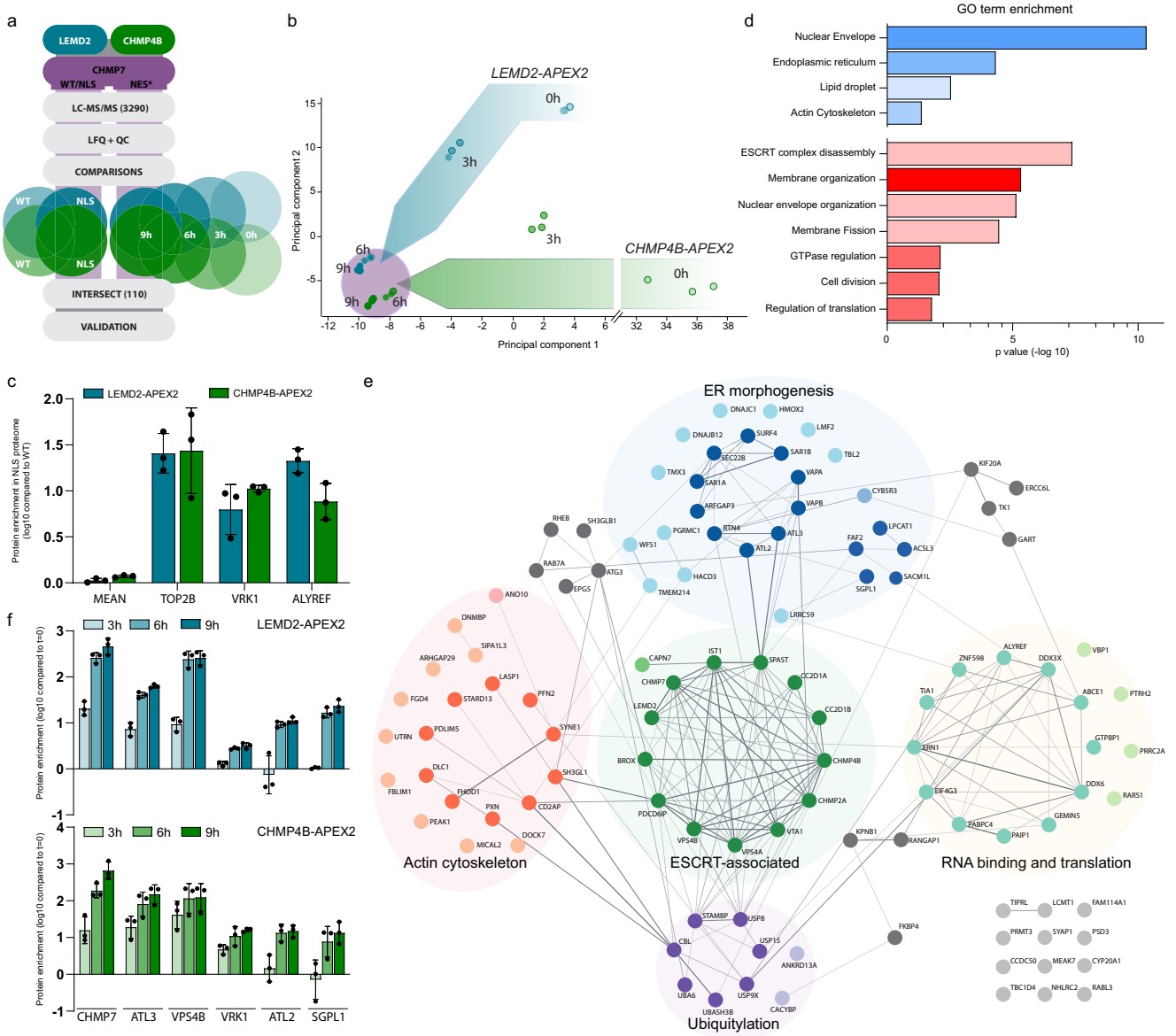

**Fig. 2 | Proximity proteomics identifies factors associated with NE rupture events. a** Schematic pipeline for MS/MS data analysis. **b** Progressive overlap of LEMD2-APEX2 and CHMP4B-APEX2 proteomes upon CHMP7[NES*] induction. Proteomic comparison performed after 0, 3, 6, and 9 h of DOX-induced CHMP7[NES*] expression. Principal component analysis at indicated timepoints, with all 3 technical replicates for each. **c** Top2B, VRK1, and ALYREF are selectively enriched in nuclear CHMP7[NLS] proteomes. Log10 scale representations of enrichments in LEMD2-APEX2 (blue) and CHMP4B-APEX2 (green) proteomes of CHMP7[NLS]/CHMP7[WT] abundance. Mean enrichment of all other 108 hits used as reference (MEAN). Error bars: mean ± SD. Each dot represents the mean per technical replicate; $N = 3$. **d** GO-term annotation of the most overrepresented functional cellular

components (blue shades) and biological processes (red shades). Bars reflect Bonferroni-corrected Fischer's exact test p-values (-log10) and color scaling intensities indicate protein count. **e** Cytoscape clustering of the 110 strongly enriched proteins identified in proteomics (details in Methods), with edges based on String-DB annotations. Nodes are functionally categorized (colored and labeled). **f** Proteomics hits show distinct temporal enrichment kinetics. Enrichment (log10 scale) of indicated targets (over t = 0 h) in LEMD2-APEX2 (blue, top panel) or CHMP4B-APEX2 (green, bottom panel) proteomes at different CHMP7[NES*] induction timepoints. Error bars: mean ± SD. Each dot represents the mean per technical replicate; $N = 3$. Source data are provided as a Source Data file.

regulation of NE dynamics through its phosphorylation of BAF[44], an early chromatin-dependent responder to NE ruptures and direct binding partner of LEMD2[15]. While we did identify BAF enrichment in our proteomics, it did not meet all the stringent criteria to be included in our shortlist. The brevity of this shortlist suggests that our proteome is largely independent of the localization of the core LEMD2-CHMP7-CHMP4B machinery and highlights only minor influence by local factors such as chromatin.

We therefore focused on proteins that were strongly enriched in both the LEMD2 and CHMP4B proximity proteomes after induction of all three CHMP7 alleles. This generated a shortlist of 110 proteins –

including LEMD2, CHMP4B, and CHMP7 – that passed all quality control and enrichment criteria (Supplementary Data 1; Methods). To assess functional enrichments of this NE repairome, we determined enriched gene annotation (GO) terms (Fig. 2d). The most significant GO cellular component terms (Fig. 2d, blue bars) relate to the NE (GO:0005635) and ER (GO:0005783), indicating the coherence of our proteome. In further support, the GO term relating to lipid droplets (GO:0005811) consisted of multiple proteins involved in phospholipid synthesis and could reflect ER membrane biogenesis needed at sites of NE rupture[45]. These ER and NE dynamics GO terms were similarly reflected in the enriched GO biological processes (Fig. 2d, red bars;

GO:0061024, GO:0006998), with a strong enrichment for the ESCRT-III complex (GO:1904903) and membrane sculpting (GO:0061024, GO:0090148, GO:0051056). Next, we organized the 110 proteins (nodes) into functional clusters (Fig. 2e), connected through STRING[46,47] annotations (edges). The core cluster (ESCRT-III-associated) consisted of 14 central ESCRT-III-associated factors[14] that have all previously been implicated in NE sealing, providing an important validation of our approach. Of these, the presence of the microtubule-severing ATPase Spastin and CC2D1B[22,48] likely reflects proteomics of LEMD2-CHMP7-CHMP4B-dependent NE reformation during cell division (GO:0051301; Supplementary Datae 1)[20–22]. Interestingly, the identification of the ESCRT-III-associated protease Calpain 7 (CAPN7)[49] and deubiquitinases STAMBP/AMSH and USP8[50] suggests additional roles for these proteins in NE repair (Fig. 2e). The ER morphogenesis cluster contained multiple proteins previously associated with ESCRTs[51,52]. Finally, the actin cytoskeleton cluster contains the linker of nucleoskeleton and cytoskeleton (LINC) complex Nesprin component SYNE1, whose paralog SYNE2 has recently been implicated in ESCRT-III dependent NE repair[41]. As such, this cluster of factors may further link perinuclear actin to NE repair processes.

Finally, we exploited the temporal element of our data set to identify proteins that become enriched at different timepoints after CHMP7 induction (Fig. 1). Only 12 proteins were highly enriched already 3 h after induction of CHMP7[NES*] expression. As expected, these mainly included core ESCRT-III associated factors (CHMP7, LEMD2, VPS4B, VTA1, BROX, and CC2D1A[14,22,48,53], but also the ER morphogenesis regulator Atlastin 3 (ATL3), and LINC complex component Nesprin-1 (SYNE1) (Fig. 2f). The majority of identified proteins were strongly enriched from the 6 h post-induction timepoint onwards, coinciding with the appearance of prominent ESCRT-III foci and onset of NE ruptures (Figs. 1, 2f).

## LRRC59 associates with LEMD2 and CHMP7 at the INM and ER
Among the shortlisted proteins, the leucine-rich repeat protein LRRC59 was one of the most enriched upon assembly of the LEMD2-CHMP7 compartmentalization sensor. LRRC59 rapidly enriched to >1000-fold and >25-fold enrichment in CHMP4B and LEMD2 snapshots, respectively (Fig. 3a and b). LRRC59 is an ER[54] and INM[55] resident transmembrane protein that has been suggested to cooperate with the importins KPNA1 and KPNB1[54] to regulate nuclear import[54,56,57]. We used immunoblotting to validate that LRRC59 was enriched in streptavidin pulldowns after CHMP7[NES*] induction (Supplementary Fig. 2A), and confirmed that CHMP7[NES*] expression caused a marked relocalization of endogenous LRRC59 to CHMP4B, LEMD2, and CHMP7 foci (Fig. 3c). The co-localization of LRRC59 with NE repair proteins upon activation of the LEMD2-CHMP7 sensor suggested that LRRC59 associates with the NE repair machinery. To assess the association of CHMP7 and LRRC59, we used bimolecular fluorescence complementation (BiFC) assays in which protein-protein interaction generates a fluorescent signal. We found a strong perinuclear BiFC signal upon co-transfection of CHMP7[WT]-mVC and mVN-LRRC59 fusions, consistent with an interaction between these proteins (Supplementary Fig. 2B–D). Complementary proximity ligation assays (PLA)[58] showed that endogenous LRRC59 specifically associated with endogenous CHMP7 and induced CHMP7[NES*] (Supplementary Fig. 2E, F). While much of the interaction associated with CHMP4B enrichment, not all PLA foci showed colocalization with CHMP4B foci (Supplementary Fig. 2F) suggesting the interaction could occur independently of the downstream recruitment of ESCRT-III subunits. Finally, PLA showed that endogenous LRRC59 interacted with LEMD2-mCitrine predominantly at the NE, but also in cytoplasmic foci likely associated with the ER (Fig. 3d, Supplementary Fig. 2G).

To further dissect the interaction between LRRC59 and the LEMD2-CHMP7 compartmentalization sensor, we generated a series of LRRC59 deletions (Fig. 3e, right panel) and assessed their ability to colocalise with overexpressed CHMP7[WT]. Strikingly, while deletion of

domains facing the cytosol/nucleoplasm (LRRC59[ΔLRR], LRRC59[ΔCC]) did not affect colocalization, deletion of the short (42 aa) domain facing the ER lumen (LRRC59[ΔLUM]) was sufficient to abrogate the colocalization of LRRC59 with overexpressed CHMP7[WT] (Fig. 3e), without apparent perturbation of the interaction between LEMD2 and CHMP7 (Fig. 3f). While CHMP7 lacks ER luminal domains, LEMD2 is a dual-transmembrane protein with an ER luminal domain (143 aa) of unknown function. We therefore reasoned LRRC59 recruitment could be mediated through interaction with LEMD2 in the ER lumen. Indeed, depletion of LEMD2 blunted the enrichment of LRRC59 at DOX-induced CHMP7[NES*] foci (Fig. 3g), indicative of its reliance on the presence of LEMD2. To assess interactions between the luminal domains of LRRC59 and LEMD2, we generated stable cell lines expressing LRRC59 C-terminally fused to a TurboID biotinylation enzyme, that faces the ER lumen. Streptavidin-based precipitation of biotinylated proteins (Supplementary Fig. 2H) identified LEMD2 (Fig. 3h), but not other transmembrane proteins with luminal domains (e.g., LBR) or INM-facing proteins (Lamin A/C). Collectively, these findings identify LRRC59 as an interactor of the nuclear compartmentalization sensor LEMD2-CHMP7, with interaction mediated through the ER luminal domains of LRRC59 and LEMD2.

## LRRC59 regulates LEMD2 accumulation during NE reformation
Depletion of LRRC59 in RPE1 cells resulted in elevated prevalence of nuclear invaginations and intranuclear tubules enriched in LEMD2 (Fig. 4a–c; Supplementary Fig. 3A–D). This phenotype could be rescued by an siRNA-resistant full-length LRRC59 allele (LRRC59[FL]), but not by the LRRC59[ΔLUM] allele that lacks the LEMD2-interacting ER luminal domain (Fig. 4d). The intranuclear LEMD2 tubule phenotype resembled those observed following depletion of CHMP7 or the downstream ESCRT-III subunit CHMP2A (Fig. 4c; Supplementary Fig. 3D), suggesting that LRRC59 directly affects ESCRT-III-dependent NE remodeling during mitotic exit[16,48]. Live-cell microscopy of synchronized RPE1 mCitrine-LEMD2 cells showed that LRRC59 depletion resulted in increased LEMD2 recruitment to and residence at the reforming NE (Fig. 4e and f; Supplementary video 1). These phenotypes were very similar to those observed upon CHMP7 or CHMP2A depletion (Supplementary Fig. 3E). LEMD2 enrichment did not resolve following telophase but persisted and developed into the intranuclear tubules in interphase cells (Supplementary video 2; Supplementary Fig. 3F). This was associated with nuclear dysmorphia such as an increased prevalence of nuclear herniations (Fig. 4g, Supplementary Fig. 3F daughter cell 1), suggesting that the NE reformation defects predisposed to nuclear fragility. In support of this, co-depletion of LRRC59 and Lamin B1, the nuclear lamina component that regulates nuclear shape and stiffness[1], synergistically increased the fraction of cells experiencing nuclear herniations and NE ruptures (Fig. 4g, Supplementary Fig. 3G and 3H). Together, our data argue that LRRC59 regulates the recruitment and subsequent timely dispersal of LEMD2 during NE reformation.

## LRRC59 regulates the LEMD2-CHMP7 compartmentalization sensor during interphase NE repair
Work from our lab and others has previously highlighted the central role for LEMD2 and CHMP7 as a compartmentalization sensor that interact at NE lesions to recruit ESCRT-III and mediate membrane sealing[17,18,20–23]. Considering our observations during mitotic NE reformation, we wondered whether LRRC59 controlled the recruitment of these factors to NE lesions during interphase as well. We found that LRRC59 depletion resulted in the hyper-accumulation and increased retention time of LEMD2 at NE lesions (Fig. 5a and b; Supplementary video 3). Importantly, this was also accompanied by increased retention of CHMP7 (Fig. 5c; Supplementary Fig. 4A; Supplementary Video 4), arguing that LRRC59 controls the association of LEMD2 and CHMP7 at these sites.

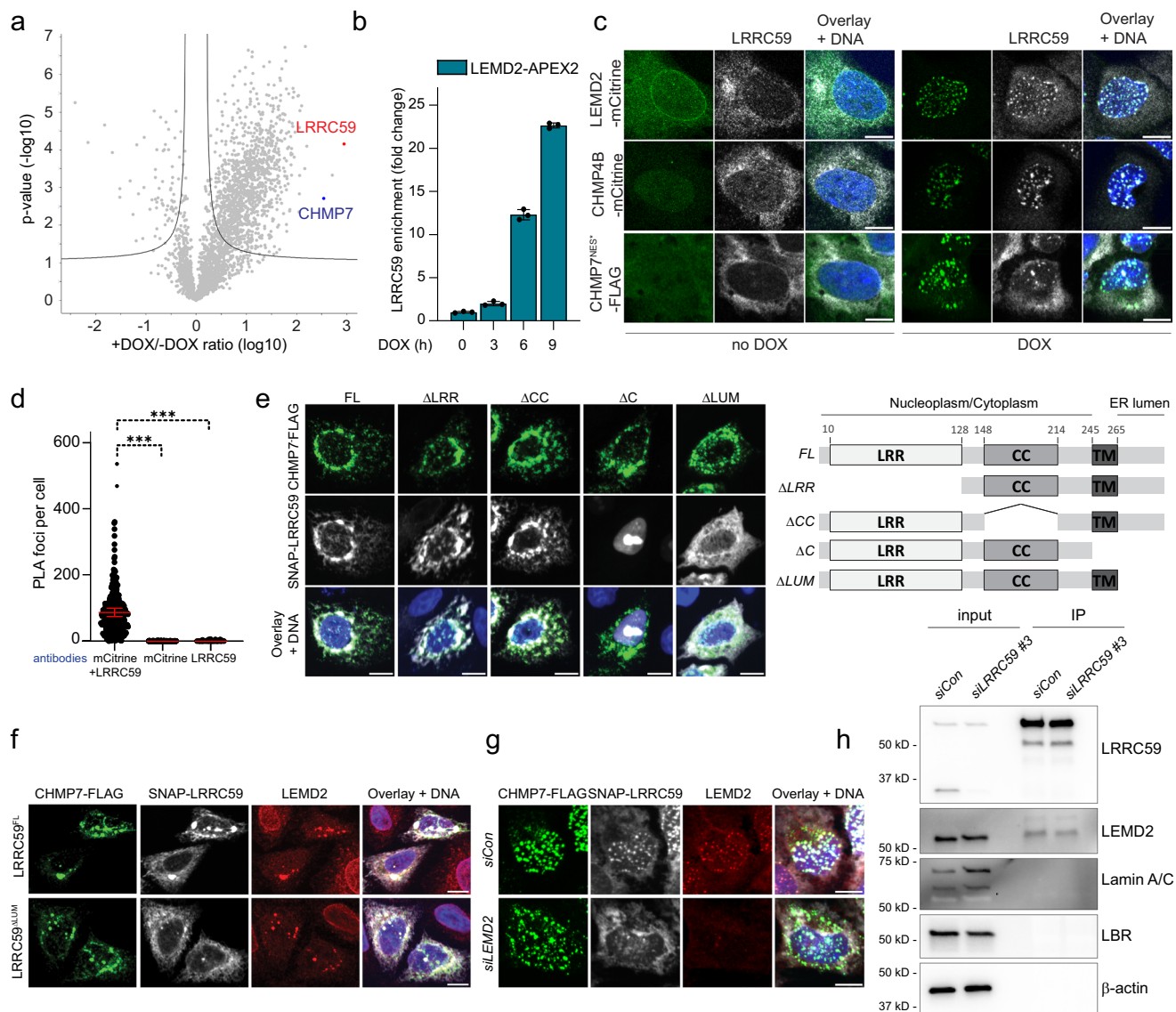

**Fig. 3 | LRRC59 is a CHMP7- and LEMD2-associated factor. a** LRRC59 enriches in RPE1 CHMP4B-APEX2 proteomes after CHMP7[WT] induction. Quantification shows protein abundance (log10 values) of +DOX/-DOX ratios. LRRC59 (red), CHMP7 (blue) highlighted. Y-axis shows adjusted p-value threshold (two-tailed Student's t-test). **b** LRRC59 proteomics enrichment following CHMP7[NES*] induction in LEMD2-APEX2 cells (indicated timepoints) and biotin-phenol labeling. Error bars: mean ± SD; $N = 3$, technical replicates. **c** Endogenous LRRC59 relocalises from the ER and INM to LEMD2/CHMP4B foci after CHMP7[NES*] expression. Representative confocal images of RPE1 LEMD2-APEX2-mCitrine or CHMP4B-APEX2-mCitrine cells following CHMP7[NES*] induction ( + DOX). Cells stained for LRRC59, DNA (Hoechst). $N = 3$. **d** Endogenous LRRC59 colocalises with LEMD2-mCitrine at the INM and ER. Quantification of PLA foci in RPE1 LEMD2-APEX2-mCitrine cells with antibodies against LRRC59 and LEMD2-mCitrine as indicated. Error bars: mean ± SEM, three independent experiments; $n = 416$ (mCitrine + LRRC59), 602 (mCitrine), 393 cells (LRRC59). mCitrine + LRRC59 vs mCitrine or vs LRRC59, ***$P = 0.0003$. One-way ANOVA with Dunnett's test. **e** Depletion of LRRC59 luminal domain abrogates colocalization with CHMP7. Left panel, representative confocal images of HeLaK

cells co-transfected with CHMP7[WT] and LRRC59 deletion constructs, stained for SNAP, FLAG, and DNA (Hoechst). $N = 3$. Right panel, schematic of hsLRRC59 domains and specific deletions used. FL: Full-length, LRR: Leucine-rich repeat, CC: Coiled-coil, LUM: ER luminal domain, TM: Transmembrane domain. **f** Depletion of the LRRC59 ER luminal domain does not affect LEMD2-CHMP7 interaction. Representative confocal images of HeLaK cells co-transfected with CHMP7[WT] and SNAP-LRRC59. Cells were stained for SNAP, FLAG, LEMD2, and DNA (Hoechst). $N = 3$. **g** LEMD2 mediates LRRC59 recruitment to CHMP7 foci. Representative confocal images of RPE1 SNAP-LRRC59[FL] CHMP7[NES*] cells, after indicated siRNA-treatments, and CHMP7[NES*] induction (DOX). Cells were stained for SNAP, FLAG, LEMD2, and DNA (Hoechst). $N = 3$. **h** Interaction of LRRC59 and LEMD2 is mediated through their ER luminal domains. Western blot of input and pulldown (IP) fractions from indicated RPE1 LRRC59-TurboID cells after biotin labeling, detected using specific antibodies against LRRC59, LEMD2, Lamin A/C, LBR, with β-actin serving as a loading control. $N = 3$. All scale bars, 10 μm. Source data are provided as a Source Data file.

Balanced recruitment of LEMD2 and CHMP7 is critical to NE repair, and we have previously shown that its imbalance causes failure to repair ruptured MN, promoting DNA damage and micronuclear catastrophe[18]. We assessed the effects of LRRC59 depletion-mediated deregulation of LEMD2 and CHMP7 recruitment upon NE rupture of the PN. This showed that LRRC59 depletion caused a significant increase in NE rupture frequency (Supplementary Fig. 4B and C). We

next used automated image analysis to measure mCherry-NLS nuclear translocation kinetics during NE rupture-repair cycles[59]. While mCherry-NLS nuclear re-entry (indicative of ensuing NE repair) showed slightly altered kinetics (Supplementary Fig. 4D), the overall repair halftime was not significantly affected (Supplementary Fig. 4E; Supplementary Video 5). mCherry-NLS is a reliable marker for the detection of NE ruptures, but it is debated whether its nuclear re-entry

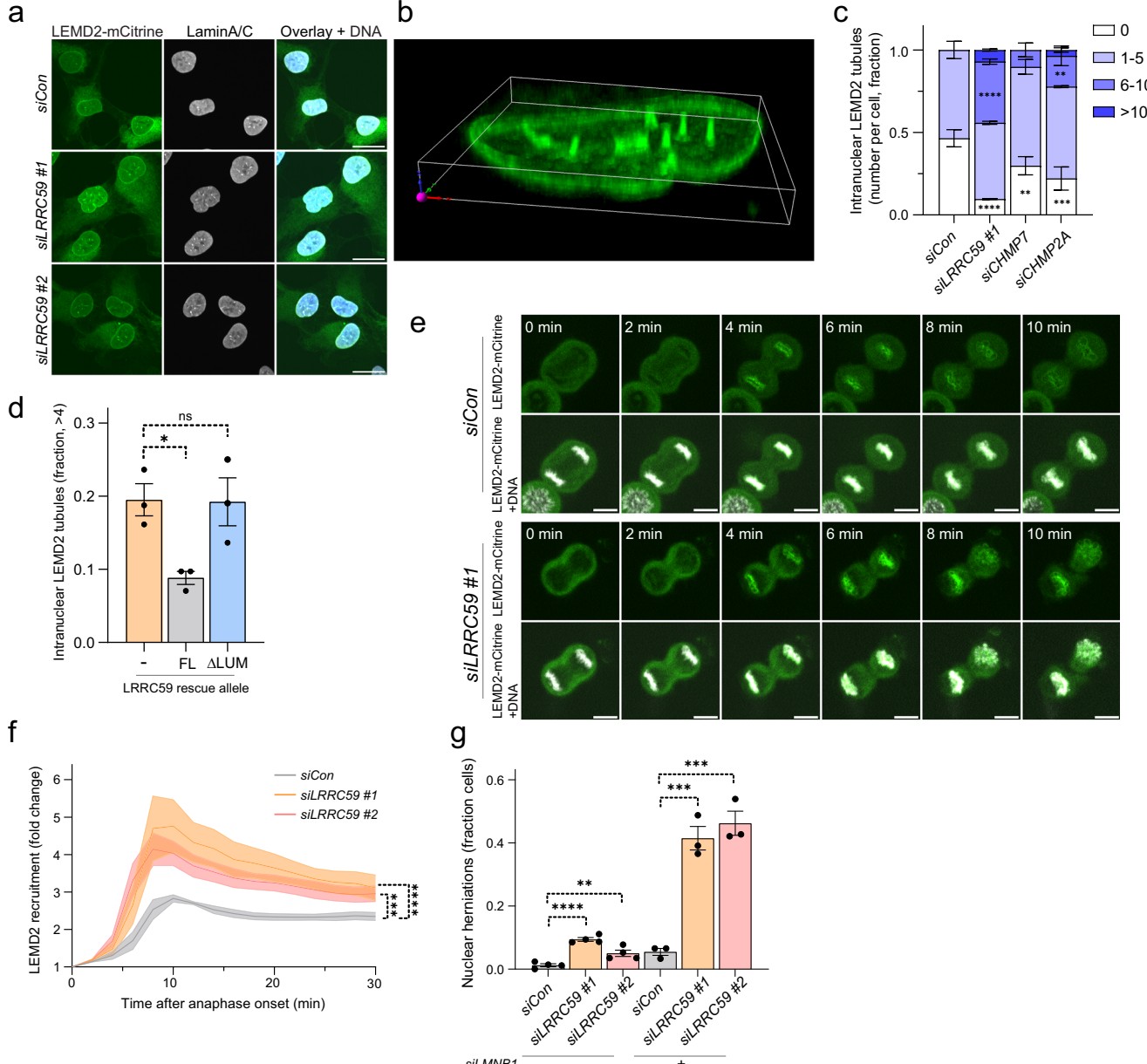

**Fig. 4 | LRRC59 regulates LEMD2 dynamics during NE reformation.**
**a** Representative confocal images of LRRC59-depleted interphase RPE1 LEMD2-APEX2-mCitrine cells displaying nuclear invaginations and intranuclear tubules. DNA counterstain (Hoechst). Scale bars, 20 μm; $N = 3$. **b** Three-dimensional reconstruction shows nuclear LEMD2 tubules post-LRRC59 depletion.
**c** Quantification of intranuclear tubules (Supplementary Fig. 3D). Error bars: mean ± SEM, 3 independent experiments, $n = 198$ (siCon); 156 (siLRRC59 #1); siCHMP7; 132 (siCHMP7); 191 cells (siCHMP2A). 0: siCon vs siLRRC59 #1 (****$P < 0.0001$), vs siCHMP7 (**$P = 0.0079$), vs siCHMP2A (***$P = 0.0001$). 6-10: siCon vs siLRRC59 #1 (****$P < 0.0001$), vs siCHMP2A (**$P = 0.0029$). Two-way ANOVA with Dunnett's test. **d** LEMD2 tubule phenotypes are rescued by siRNA-resistant LRRC59FL but not LRRC59ΔLUM. Bar graph shows cell fraction containing intranuclear LEMD2 tubules ( > 4) after endogenous LRRC59 depletion in siRNA #3-resistant SNAP-LRRC59 cells. Error bars: mean ± SEM, 3 independent experiments; $n = 262$ (-); 242 (LRRC59FL); 269 cells (LRRC59ΔLUM). - vs LRRC59FL, *$P = 0.0314$; - vs

LRRC59ΔLUM, NS, $P = 0.9952$. One-way ANOVA with Dunnett's test. **e**, **f** LRRC59 knockdown increases LEMD2 recruitment to the reforming NE. Stills (**e**) and fold increase (**f**) from live-cell imaging of RPE1 LEMD2-APEX2-mCitrine cells after indicated siRNAs post-anaphase onset (t = 0 min). DNA counterstain (SPY-650). Scale bars, 10 μm; $N = 3$. Error bars: mean (lines) ± SEM (bands); $n = 110$ (siCon); 64 (siLRRC59 #1); 66 cells (siLRRC59 #2). siCon vs siLRRC59 #1 (****$P < 0.0001$), vs siLRRC59 #2 (***$P = 0.0002$). One-way ANOVA of AUC with Dunnett's test.
**g** Synergistic effects on nuclear herniation frequency following LRRC59 and LMNB1 knockdown in RPE1 CHMP4B-mNeonGreen, mCherry-NLS cells. Error bars: mean ± SEM, 3 (with siLMNB1) or 4 (without siLMNB1) independent experiments. Without siLMNB1 $n = 438$ (siCon); 367 (siLRRC59 #1); 401 cells (siLRRC59 #2). siCon vs siLRRC59 #1 (****$P < 0.0001$), vs siLRRC59 #2 (**$P = 0.0097$). One-way ANOVA with Dunnett's test. With siLMNB1 $n = 425$ (siCon); 308 (siLRRC59 #1); 403 cells (siLRRC59 4). siCon vs siLRRC59 #1 (***$P = 0.0003$), vs siLRRC59 #2 (***$P = 0.0002$). One-way ANOVA with Dunnett's test. Source data are provided as a Source Data file.

reflects resealing of the nuclear membrane or rather the re-establishment of nucleocytoplasmic shuttling[60]. In support of the latter, CHMP2A depletion, which prevents membrane fission in other cellular ESCRT-III functions[61–64], showed similar nuclear nCherry-NLS restoration kinetics as control or LRRC59-depleted cells

(Supplementary Fig. 4F; Supplementary Video 5). To determine actual membrane repair, we instead monitored recruitment kinetics of CHMP4B-mNeonGreen (mNG) to sites of NE rupture. While CHMP2A depletion effectively trapped CHMP4B at NE lesions indicative of stalled membrane fission (Supplementary Fig. 4G–I), LRRC59

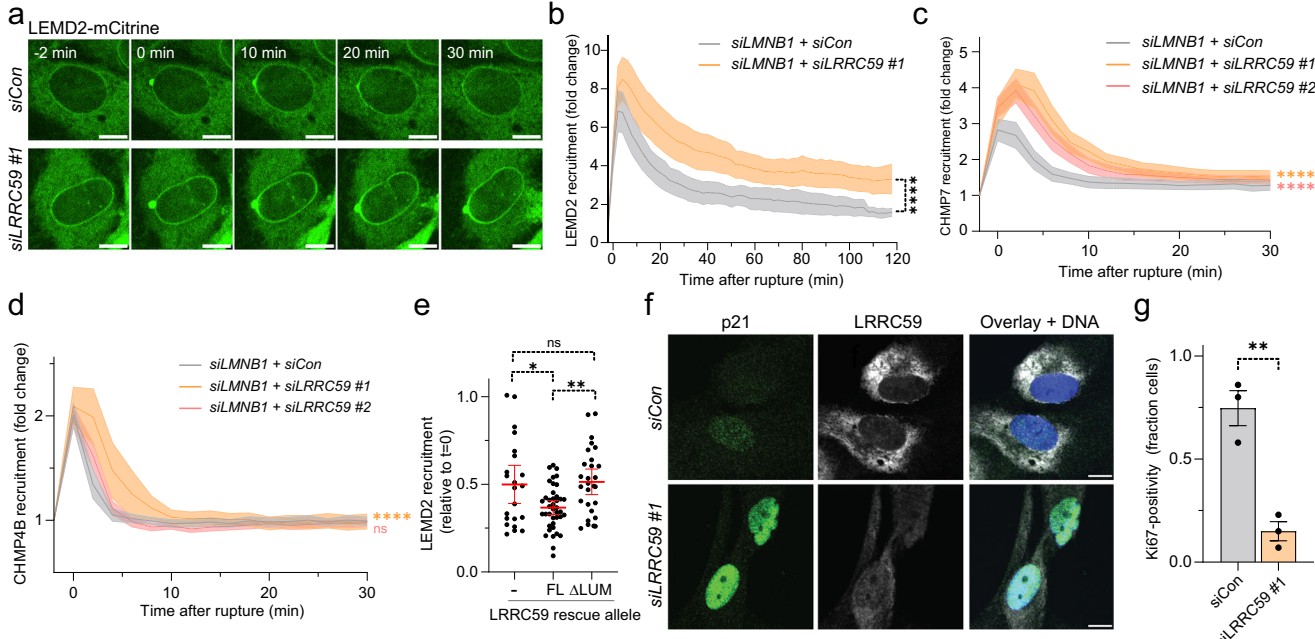

**Fig. 5 | LRRC59 regulates the LEMD2-CHMP7 compartmentalization sensor in NE rupture and repair. a** LRRC59 depletion causes hyperaccumulation of LEMD2 at NE ruptures. Stills from live-cell imaging of RPE1 LEMD2-APEX2-mCitrine cells treated with indicated siRNAs. Scale bars, 10 μm; $N = 3$. **b** Quantification of LEMD2 recruitment to NE ruptures (panel **a**). Error bars: mean (lines) ± 95% CI (bands), 3 independent experiments; $n = 35$ (siLMNB1 + siCon); 50 cells (siLMNB1 + siLRRC59 #1). siLMNB1 + siCon vs siLMNB1 + siLRRC59 #1, ****$P < 0.0001$. Two-tailed unpaired Student's t-test of AUC. **c** LRRC59 depletion increases and prolongs CHMP7 recruitment to NE ruptures. Quantification of CHMP7 recruitment in RPE1 HAeGFP-CHMP7 live-cell imaging. Error bars: mean (lines) ± 95% CI (bands), 3 independent experiments; $n = 44$ (siLMNB1 + siCon); 61 (siLMNB1 + siLRRC59 #1); 74 cells (siLMNB1 + siLRRC59 #2). siLMNB1 + siCon vs siLMNB1 + siLRRC59 #1, ****$P < 0.0001$; siLMNB1 + siCon vs siLMNB1 + siLRRC59 #2, ****$P < 0.0001$. One-way ANOVA of AUC with Dunnett's test. **d** LRRC59 depletion has marginal effects on ESCRT-III dynamics at NE ruptures. Quantification of CHMP4B-LAP-mNG intensities at NE ruptures. Error bars: mean (lines) ± 95% CI (bands), 3 independent

experiments; $n = 53$ (siLMNB1 + siCon); 42 (siLMNB1 + siLRRC59 #1); 60 cells (siLMNB1 + siLRRC59 #2). siLMNB1 + siCon vs. siLMNB1 + siLRRC59 #1, ****$P < 0.0001$; siLMNB1 + siCon vs. siLMNB1 + siLRRC59 #2, NS, $P = 0.7344$. One-way ANOVA of AUC with Dunnett's test. **e** LEMD2 hyperaccumulation at NE ruptures is rescued by LRRC59$^{FL}$ but not LRRC59$^{ΔLUM}$. Quantification of LEMD2-APEX2-mCitrine intensities at NE ruptures one hour post-rupture (fraction of max intensity) from endogenous LRRC59-depleted RPE1 LEMD2-APEX2-mCitrine cells expressing siRNA #3-resistant SNAP-LRRC59. Error bars: mean ± 95% CI of 3 independent experiments; $n = 22$ (-); 40 (LRRC59$^{FL}$); 28 cells (LRRC59$^{ΔLUM}$). - vs. LRRC59$^{FL}$, *$P = 0.0187$; - vs. LRRC59$^{ΔLUM}$, NS, $P = 0.9547$; LRRC59$^{FL}$ vs. LRRC59$^{ΔLUM}$, **$P = 0.0038$. One-way ANOVA with Tukey's test. **f, g** LRRC59 depletion causes cell stress and reduced proliferation. **f,** representative confocal images of control and LRRC59-depleted cells stained for p21, LRRC59, and DNA (Hoechst). Scale bars, 10 μm; $N = 3$. **g,** as **f** but quantified for Ki67 (fraction of Ki67-positive cells). Error bars: mean ± SEM of 3 independent experiments; $n = 261$ (siCon); 207 cells (siLRRC59 #1). **$P = 0.0035$, two-tailed unpaired Student's t-test. Source data are provided as a Source Data file.

depletion showed normal kinetics of CHMP4B recruitment and disassembly (Fig. 5d; Supplementary Fig. 4H and I; Supplementary Video 5). These experiments argue that NE sealing is executed normally in the absence of LRRC59 and rather points towards a defect in controlling the accumulation and resolution of the LEMD2-CHMP7 compartmentalization sensor. This regulation requires the association of LRRC59 with LEMD2, as a defect in LEMD2 dynamics could not be rescued by an siRNA-resistant LRRC59$^{ΔLUM}$ allele (Fig. 5e).

Our previous work has shown that compromised nuclear integrity culminates in increased cell stress, as exemplified by upregulation of p21[22]. We found that LRRC59 depletion caused a similar increase in nuclear p21 levels, and this was associated with reduced levels of the cell proliferation markers Ki67 and CCNB1 (Fig. 5f and g, Supplementary Fig. 5A–C). Therefore, LRRC59-mediated control of the nuclear compartmentalization sensor is essential to ensure long-term cell fitness.

## LRRC59, KPNB1, and XPO1 guide the assembly of the compartmentalization sensor

Disruption of Exportin 1 (XPO1)-mediated export following NE rupture results in nuclear accumulation of CHMP7, which triggers assembly of LEMD2-CHMP7 complexes and recruitment of downstream ESCRT-III subunits[17,18]. We have shown that this process could be mimicked by artificially targeting CHMP7 to the nucleus (by mutation of its NES or

fusion to an NLS)[18] (Fig. 1). We noticed that LEMD2 accumulated into larger NE foci specifically in CHMP7$^{NES*}$ as compared to CHMP7$^{NLS}$ cells (Fig. 6a). We therefore hypothesized that XPO1 association with CHMP7 not only controls its nuclear export but could also modulate its association with LEMD2. To test this, we incubated cells with Leptomycin B (LMB) that covalently binds to XPO1 and thereby disrupts its physical association with cargos[18,65]. Strikingly, LMB treatment of CHMP7$^{NLS}$ expressing cells resulted in accumulation of LEMD2 into larger foci, resembling those observed in CHMP7$^{NES*}$ cells (Fig. 6b). We did not observe an effect on foci size upon LMB treatment of cells expressing the CHMP7$^{NES*}$, which is already impaired in binding to XPO1 (Fig. 6c; Supplementary Fig. 6A). This showed that the LMB effect on LEMD2 accumulation was mediated directly through association of XPO1 with CHMP7 rather than through another XPO1 cargo.

The above results were reminiscent of our observations at NE ruptures following LRRC59 depletion (Fig. 5a and b). Indeed, we found that depletion of LRRC59 resulted in increased LEMD2 foci size upon induction of either CHMP7$^{NLS}$ or CHMP7$^{NES*}$ (Fig. 6d and e; Supplementary Fig. 6B), indicating that LRRC59 regulates LEMD2 independently of the XPO1-CHMP7 axis. LRRC59 has previously been suggested to associate with the importin KPNB1[54], raising the possibility that they together constitute a complementary regulatory node controlling LEMD2 accumulation. In support of this, KPNB1 accumulated in the NE repairome with similar kinetics as LRRC59 (Fig. 6f),

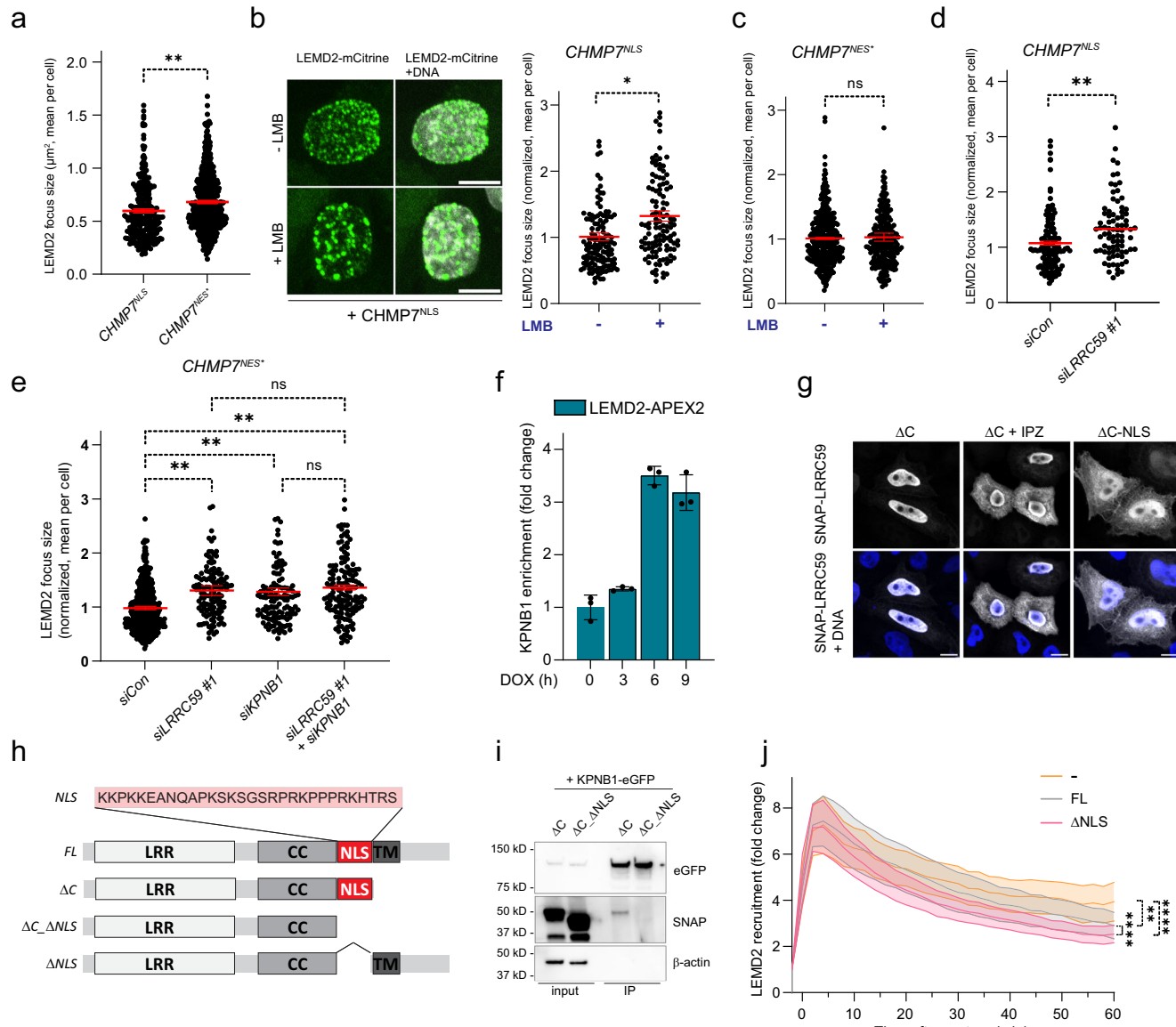

**Fig. 6 | LRRC59 interaction with KPNB1 regulates LEMD2 accumulation at NE ruptures. a** LEMD2 accumulates into larger foci upon CHMP7$^{NES*}$ versus CHMP7$^{NLS}$ overexpression. Live-cell imaging of RPE1 LEMD2-APEX2-mCitrine cells and LEMD2 foci size quantification after CHMP7 induction. Error bars: mean ± SEM, 6 independent experiments, dots represents mean foci size/cell; $n$ = 564 (CHMP7$^{NES*}$); 263 cells (CHMP7$^{NLS}$). **$P$ = 0.0083, two-tailed unpaired Student's t-test. **b** Inhibiting XPO1 increases LEMD2 foci size upon CHMP7$^{NLS}$ induction. Stills from live-cell imaging of RPE1 LEMD2-APEX2-mCitrine cells treated with LMB (left panel). DNA counterstain (SPY-650). Scale bars, 10 μm; $N$ = 3. Quantifications of LEMD2 foci size (right panel). Error bars: mean ± SEM, 3 independent experiments, dots represent mean foci size/cell; $n$ = 138 (-LMB); 117 cells (+LMB). *$P$ = 0.0306, two-tailed unpaired Student's t-test. **c** Blocking XPO1 binding does not affect LEMD2 foci size in CHMP7$^{NES*}$ cells. Quantification as in (**b**). $n$ = 359 (-LMB); 248 cells (+LMB). NS, $P$ = 0.7836, two-tailed unpaired Student's t-test. **d, e** LRRC59 and KPNB1 depletion regulate LEMD2 foci size upon CHMP7$^{NLS}$ or CHMP7$^{NES*}$ overexpression. **d** quantification of LEMD2 foci size following indicated treatments and CHMP7$^{NLS}$ induction. Error bars: mean ± SEM, 3 independent experiments, dots represent mean foci size/cell; $n$ = 125 (siCon); 82 cells (siLRRC59 #1). **$P$ = 0.0025, two-tailed unpaired Student's t-test. **e** quantification as in (**d**) but now after CHMP7$^{NES*}$ expression. Error bars: mean ± SEM, 3 independent experiments, with dots representing the mean foci size/cell; $n$ = 421 (siCon); 126 (siLRRC59); 123 (siKPNB1); 143

cells (siLRRC59 + siKPNB1). siCon vs siLRRC59 (**$P$ = 0.0048), vs siKPNB1 (**$P$ = 0.0077), vs siLRRC59 + siKPNB1 (**$P$ = 0.0015); siLRRC59 vs siKPNB1 ($P$ = 0.9159), vs siLRRC59 + siKPNB1 ($P$ = 0.8093). One-way ANOVA with Tukey's multiple comparison test. **f** KPNB1 proteomics enrichment post-CHMP7$^{NES*}$ induction in LEMD2-APEX2-expressing cells DOX-treated as indicated to induce CHMP7$^{NES*}$ expression (0 h to 9 h). Error bars: mean ± SD. Dot represents mean/ technical replicate; $N$ = 3. **g, h** LRRC59 contains a bona fide NLS. **g** Representative confocal images of RPE1 cells transfected with SNAP-LRRC59$^{ΔC}$ or SNAP-LRRC59$^{ΔC\_ΔNLS}$ and treated as indicated. Cells were stained for SNAP and DNA (Hoechst). Scale bars, 10 μm; $N$ = 3. **h** Schematic of hsLRRC59 with NLS and used deletions indicated. FL: Full-length, LRR: Leucine-rich repeat, CC: Coiled-coil, TM: Transmembrane. **i** LRRC59 interacts with KPNB1 via its NLS. Co-immunoprecipitation of HEK293T cells cotransfected with KPNB1-eGFP and SNAP-LRRC59. Representative of 3 experiments. **j** LRRC59$^{ΔNLS}$ accelerates LEMD2 dissolution at NE ruptures. Quantification of LEMD2-APEX2-mCitrine (fold increase) at NE ruptures upon endogenous LRRC59 depletion with or without expression of siRNA #3-resistant SNAP-LRRC59. Error bars: mean (lines) ± 95% CI (bands), 3 independent experiments; $n$ = 27 (-); 44 (LRRC59$^{FL}$); 43 cells (LRRC59$^{ΔNLS}$). - vs. LRRC59$^{FL}$, **$P$ = 0.0042; - vs. LRRC59$^{ΔNLS}$, ****$P$ < 0.0001; LRRC59$^{FL}$ vs. LRRC59$^{ΔNLS}$, ****$P$ < 0.0001. One-way ANOVA of AUC with Tukey's test. Source data are provided as a Source Data file.

although enrichment was less pronounced (3-fold vs 25-fold) (Fig. 3b). Furthermore, we found that a soluble LRRC59$^{\Delta C}$ truncation was exclusively nuclear (Figs. 3e and 6g) and that this enrichment was blocked upon inhibition of importins by Importazole (IPZ) treatment[66] (Fig. 6g, Supplementary Fig. 6C), arguing that LRRC59 is actively regulated by KPNB1. Using a series of microdeletions, we identified a short sequence element juxtapositioned to the TM and LUM domains that functions as a bona fide NLS in LRRC59 (Fig. 6h, Supplementary Fig. 6C). Finally, co-immunoprecipitation experiments showed that LRRC59 truncations retaining the NLS sequence (LRRC59$^{\Delta C}$) physically associated with GFP-KPNB1, while additional deletion of this NLS (LRRC59$^{\Delta C\_\Delta NLS}$) abrogated this interaction (Fig. 6i). Importantly, depletion of KPNB1 phenocopied LRRC59 knockdown with regard to enlargement of LEMD2 foci after CHMP7$^{NES*}$ induction (Fig. 6e; Supplementary Fig. 6B). As co-depletion of KPNB1 and LRRC59 did not further accentuate this phenotype (Fig. 6e), this further argued that LRRC59 and KPNB1 work together to control LEMD2 accumulation.

To determine the importance of this interaction, we assessed the ability of an LRRC59 fusion lacking the NLS (LRRC59$^{\Delta NLS}$) to rescue the elevated and persistent LEMD2 recruitment to NE ruptures after LRRC59 depletion. In contrast to LRRC59$^{\Delta LUM}$ (Fig. 5e), the LRRC59$^{\Delta NLS}$ fully rescued this phenotype (Fig. 6j). In fact, LRRC59$^{\Delta NLS}$ accelerated the resolution of LEMD2 recruitment compared to siRNA-resistant LRRC59$^{FL}$, suggesting that dissociation of LRRC59 from importins enhances its ability to resolve LEMD2 foci (Fig. 6j). As the NLS is directly juxtaposed to the transmembrane and luminal domains of LRRC59, we reasoned that binding of importins to the LRRC59 NLS could sterically interfere with the interaction of LRRC59 with LEMD2 through its LUM domain. To test this, we performed molecular modeling of LRRC59 in association with LEMD2 or importins. This molecular modeling nicely recapitulated our experimental data (Fig. 3e–h) of an interaction between the ER luminal regions of LRRC59 and LEMD2 (Supplementary Fig. 7A top panels). Similarly, these models confirmed the interaction of the LRRC59 NLS (Fig. 6g–j) with importins, consisting of KPNB1 in conjunction with its NLS-binding partner KPNA2, the highest enriched importin α in our proteomics[65] (Supplementary Fig. 7A, bottom left). However, our attempts to model the quaternary LRRC59-LEMD2-KPNB1-KPNA2 complex resulted in multiple steric clashes (Supplementary Fig. 7B). These sites of steric interference stem from the extended TM α helices in LEMD2 (extending 30–40 Å into the nucleoplasm) clashing with the wide diameter of the LRRC59-associated importins ( > 50 Å diameter), further complicated by the proximity of the C-terminal winged-helix of LEMD2. As such, these models combined with our cell biology experiments suggest that importins could directly interfere with the ability of LRRC59 to bind to LEMD2. This is particularly intriguing as the RAN gradient is perturbed at sites of NE ruptures, preventing local dissociation of KPNB1 from LRRC59.

Next, we sought to assess whether the LRRC59 and KPNB1 node cooperated with the XPO1 node during LEMD2-CHMP7 regulation during interphase NE ruptures, a physiologically relevant model. While depletion of LRRC59 mainly resulted in LEMD2 accumulation at NE lesions (Figs. 5a, 5b, 7a and 7c, Supplementary Fig. 6D and 6E), XPO1 inhibition occasionally induced a limited spread of LEMD2 from NE lesions in a fraction of ruptured cells (Fig. 7a and c, Supplementary Fig. 6D and 6E; Supplementary Video 6). Intriguingly, simultaneous perturbation of the LRRC59 and XPO1 nodes synergized to cause progressive spread of endogenous or LEMD2-mCitrine from the initial NE lesion (Fig. 7a and c, Supplementary Fig. 6E and F). This striking phenotype developed throughout the nucleus within few hours (Fig. 7a and c, Supplementary Fig. 6F), with a maximum spreading rate of $3.30 +/- 1.10$ µm²/min (Supplementary Fig. 6F, Supplementary Video 6). Together, these observations uncover a critical role of the LRRC59 and XPO1 nodes to temporally and spatially restrict LEMD2-CHMP7 accumulation to sites of rupture.

As LEMD2-CHMP7 interaction is the critical trigger for nucleation of CHMP4B filaments[18,22], we assessed whether the progressive spread of LEMD2 across the NE was accompanied by CHMP4B nucleation. Monitoring CHMP4B dynamics upon LRCC59 depletion and XPO1 inhibition by LMB showed a prominent wave of CHMP4B foci spreading progressively from the initial NE lesion (Fig. 7b and d; Supplementary Video 7), likely occurring at the leading edge of the spreading LEMD2. In contrast to LEMD2, these CHMP4B accumulations were transient (Fig. 7b; Supplementary Video 7), consistent with our observations that turnover of the downstream ESCRT-III machinery is still functional in the absence of LRRC59 (Fig. 5d, Supplementary Fig. 4H). Depletion of KPNB1 in concert with LMB resulted in a similar wave of CHMP4B accumulations (Fig. 7e). We could suppress the KPNB1 depletion effects by overexpression of LRRC59$^{FL}$ (Fig. 7e, Supplementary Fig. 6G and 6H). These data are consistent with a dual role of KPNB1 in localizing LRRC59 to the nucleus (Fig. 6g) and in antagonizing its interaction with LEMD2 at sites of NE ruptures (Fig. 6j and Supplementary Fig. 7) through persistent association of KPNB1 with LRRC59.

Previously, we have found that uncontrolled formation of ESCRT-III foci resulted in locally elevated DNA torsional stress, as LEMD2 bridges ESCRT-III and chromatin interactions[18]. We therefore assessed whether the spreading and accumulation of LEMD2 and CHMP4B upon concerted deregulation of LRRC59-KPNB1 and CHMP7-XPO1 was accompanied by increased torsional stress on chromatin. Indeed, we found a strong enrichment of the DNA torsional stress marker topoisomerase IIb (Top2B) into foci selectively within the nuclear region affected by LEMD2 spreading (Fig. 7f). These observations argue for a critical role for the LRRC59-KPNB1 and XPO1 nodes in curtailing LEMD2-CHMP7 function at the NE.

Together our data identify a key regulatory pathway at NE lesions, centered around the LRRC59-KPNB1-XPO1 axis. Our data show that this axis controls LEMD2-CHMP7 interaction and argue that importins and exportins play a role in the function of the compartmentalization sensor beyond the localization of its constituents, LEMD2 and CHMP7.

## LRRC59 controls micronuclear fate

The spread of LEMD2 and the accompanying wave of CHMP4B foci across the NE surface were highly reminiscent of our previous observations in rupturing MN[18]. Intriguingly, it has been well-established that MN frequently experience major defects in nucleocytoplasmic shuttling, as well as KPNB1 and XPO1 levels[18,67,68]. We therefore asked whether LRRC59 was recruited to rupturing MN as well and could equally affect their NE repair. Quantification of SNAP-LRRC59 levels at ruptured MN in stable RPE1 and HeLaK cell lines showed that LRRC59 indeed enriched at ruptured MN (Fig. 8a). Using live cell microscopy, we found that SNAP-LRRC59 accumulated rapidly and persistently to MN within minutes of rupture (Fig. 8b and c; Supplementary Videos 8 and 9, Supplementary Fig. 8A and 8B). Importantly, while LRRC59$^{FL}$ and LRRC59$^{\Delta NLS}$ strongly enriched at ruptured MN, this was far less pronounced for LRRC59$^{\Delta LUM}$ (Fig. 8d, Supplementary Fig. 8C) consistent with its inability to interact with LEMD2 (Fig. 3e, f). Finally, PLA experiments corroborated that endogenous LRRC59 and CHMP7 could associate at ruptured MN (Fig. 8e).

Noting that LRRC59 depletion led to increased LEMD2 levels at NE lesions in PN, we speculated the same could happen during MN ruptures. Indeed, LRRC59 depletion resulted in accelerated accumulation and higher levels of LEMD2-mCitrine recruitment to MN upon rupture (Fig. 8f and g; Supplementary Fig. 8D; Supplementary Video 10).

We have previously shown that the imbalance between LEMD2 and CHMP7 levels at MN precludes their successful repair and that a partial reduction of CHMP7 levels results in increased success of repair[18]. As LRRC59 depletion caused increased recruitment of LEMD2 to NE lesions of MN (Figs. 5a, 5b and 8f, g), we hypothesized this could contribute to restore the balance between LEMD2 and CHMP7. To assess this, we used live-cell microscopy and scored the frequency with

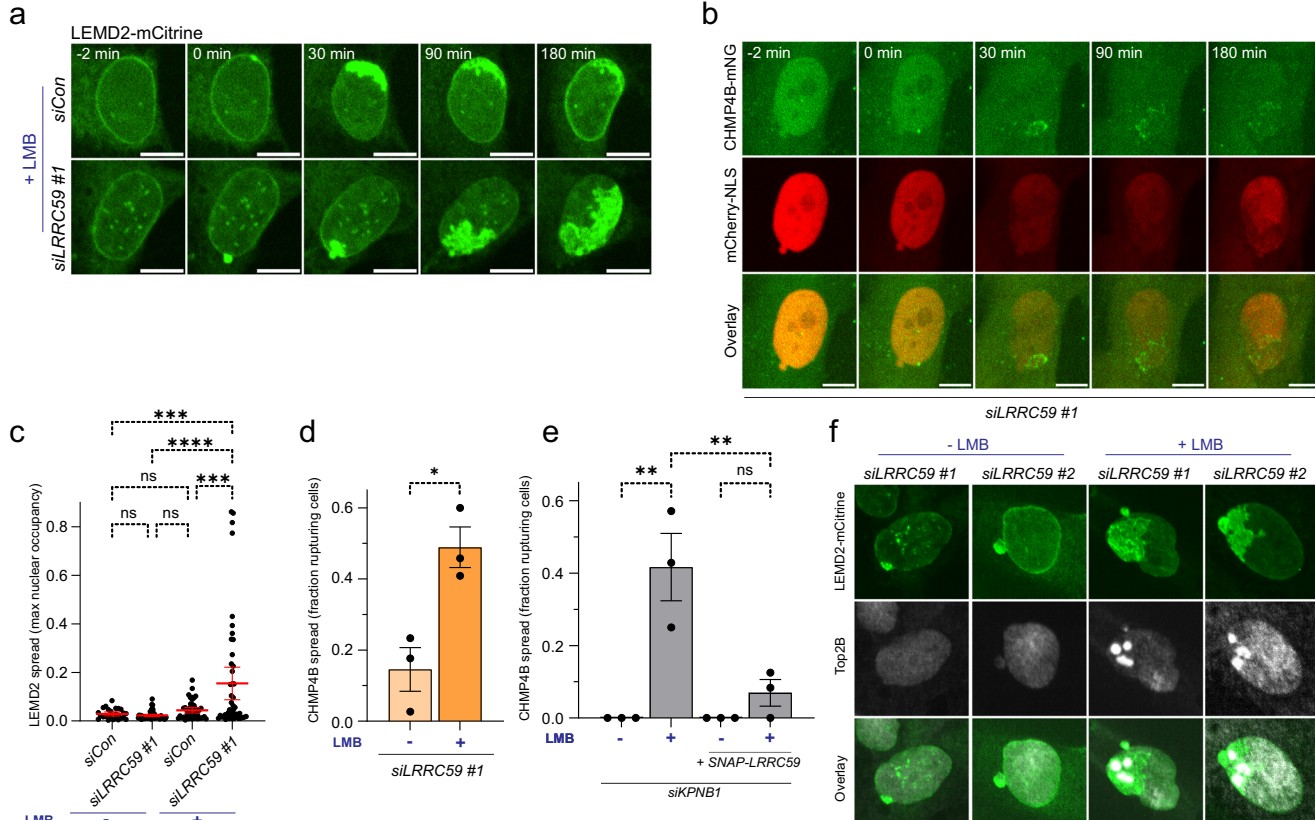

**Fig. 7 | LRRC59-KPNB1 and XPO1 constitute complementary regulatory nodes that restrain the LEMD2-CHMP7 compartmentalization sensor. a, c** Simultaneous perturbation of LRRC59-KPNB1 and XPO1 triggers extensive LEMD2 spread from NE ruptures. **a** Stills of live-cell imaging of RPE1 LEMD2-APEX2-mCitrine cells with indicated treatments. Scale bars, 10 μm; *N* = 3. **c** Quantification of maximal nuclear occupancy of LEMD2 after NE rupture. Error bars: mean ± 95% CI, 3 independent experiments, dots represent individual cells; *n* = 26 (siCon, -LMB); 47 (siCon, +LMB); 46 (siLRRC59 #1, -LMB); 49 cells (siLRRC59 #1, +LMB). siCon -LMB vs LRRC59 #1 -LMB, NS, *P* = 0.9973; siCon -LMB vs siCon +LMB, NS, *P* = 0.9624; siCon -LMB vs siLRRC59 #1 + LMB; ***P* = 0.0005; siCon +LMB vs siLRRC59 #1 -LMB, NS, *P* = 0.8521; siLRRC59 #1 -LMB vs siLRRC59 #1 + LMB, ****P* < 0.0001; siCon +LMB vs siLRRC59 #1 + LMB, ***P* = 0.0003. One-way ANOVA with Tukey's test. **b** Wave of CHMP4B spread induced by combined LRRC59-KPNB1 and XPO1 disruption. Stills from live-cell imaging of RPE1 CHMP4B-LAP-mNG mCherry-NLS cells treated with siLRRC59 #1 + siLMNB1 and LMB. Scale bars, 10 μm; *N* = 3. **d** Quantification of fraction of nuclei undergoing CHMP4B spread after NE rupture. Error bars: mean ± SEM, 3 independent experiments, dots represent mean/experiment; *n* = 84

(siLRRC59 #1, -LMB); 76 cells (siLRRC59 #1, +LMB). *P* = 0.015, two-tailed unpaired Student's t-test. **e** KPNB1 depletion-induced CHMP4B spread is suppressed by SNAP-LRRC59 overexpression. Analysis of live-cell imaging of RPE1 CHMP4B-LAP-mNG or RPE1 CHMP4B-LAP-mNG SNAP-LRRC59 cells with indicated treatments. Quantification of fraction of nuclei undergoing CHMP4B spread after NE rupture, Error bars: mean ± SEM, 3 independent experiments, dots represent mean/experiment; RPE1 CHMP4B-LAP-mNG cells, *n* = 17 (siKPNB1, -LMB); 23 cells (siKPNB1, +LMB). RPE1 CHMP4B-LAP-mNG LRRC59-SNAP cells, *n* = 8 (siKPNB1, -LMB); 21 cells (siKPNB1, +LMB). CHMP4B-LAP-mNG cells, siKPNB1 -LMB vs siKPNB1 -LMB, ***P* = 0.0016; CHMP4B-LAP-mNG LRRC59-SNAP cells, siKPNB1 -LMB vs siKPNB1 -LMB, NS, *P* = 0.7632; CHMP4B-LAP-mNG cells, siKPNB1 +LMB vs CHMP4B-LAP-mNG LRRC59-SNAP cells, siKPNB1 +LMB, ***P* = 0.0052. One-way ANOVA with Tukey's test. **f** Perturbed LEMD2 spreading induces DNA torsional stress. Representative confocal images of RPE1 LEMD2-APEX2-mCitrine cells with indicated treatments, stained for topoisomerase IIb (Top2B). Scale bars, 10 μm; *N* = 3. Source data are provided as a Source Data file.

which ruptured MN underwent successful recompartmentalization (as assessed by mCherry-NLS re-accumulation). These experiments showed significantly increased repair success rates following LRRC59 depletion (Fig. 8h). Conversely, overexpression of LRRC59[FL] from a stably integrated doxycycline (DOX)-inducible transgene for 24 h or 48 h significantly reduced the rate of MN repair (Fig. 8i). Together, these data suggest a direct role for LRRC59 in balancing the LEMD2-CHMP7 interaction compatible with repair of a subpopulation of MN.

Finally, we assessed the fate of the fraction of ruptured MNs that did not repair in the absence of LRRC59, but rather progressively accumulated LEMD2 across the entire MN – very similar to our observations in PN (Fig. 7a and c, Supplementary Fig. 6E and 5F). We observed frequent DNA torsional stress in ruptured MN (Supplementary Fig. 8E). LRRC59 depletion increased accumulation of the DNA damage marker γH2Ax as well as the fraction of γH2Ax-positive ruptured MNs (Fig. 8j; Supplementary Fig. 8F). Together these data argue that LRRC59, LEMD2, and CHMP7 together control repair of MN and

accumulation of DNA damage in ruptured MN, a major pathophysiological event associated with cancer progression[8,29–32,34,36,37].

## Discussion

Research over the last years has highlighted the high prevalence of transient and persistent NE ruptures under physiological and pathophysiological conditions[1,3,8,69]. This study provides two major steps forward in our understanding of the regulation of the cellular responses to NE ruptures. Firstly, where previous studies have relied on genetics screening approaches to find regulators on NE integrity[70,71], we provide a comprehensive proteome associated with sites of NE repair. Secondly, we uncover a key layer of regulation of NE repair mediated through concerted action of LRRC59 and nuclear transporters to restrain the LEMD2-CHMP7 compartmentalization sensor function to NE lesions.

Our convergent LEMD2-CHMP7-CHMP4B-based NE repair proteomics yields multiple leads into NE rupture-associated processes. Even though the use of transgenic fusions and inducible

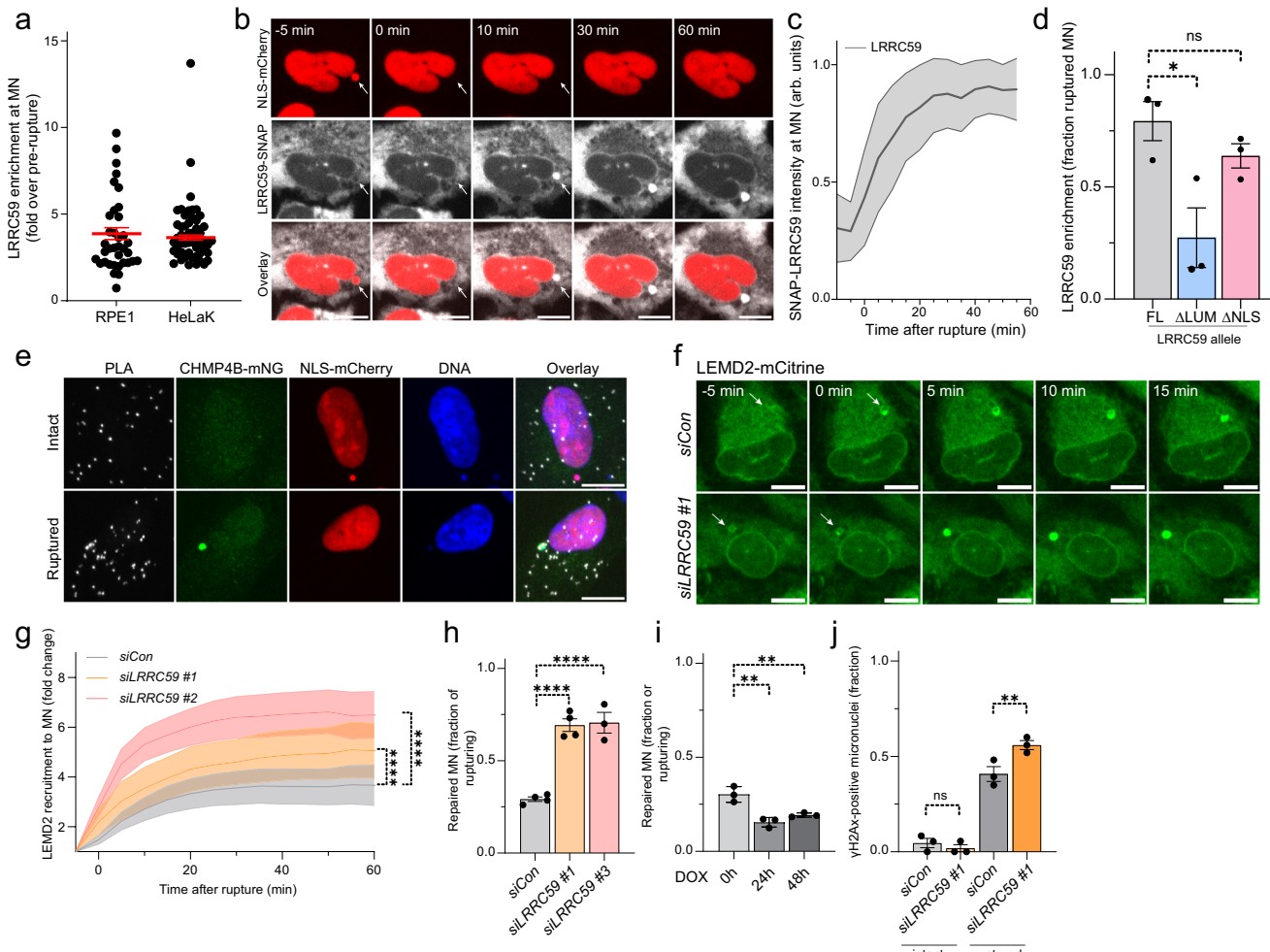

**Fig. 8 | LRRC59 controls micronuclear fate. a** LRRC59 accumulates at ruptured MN in RPE1 and HeLa K cells. Quantifications from live-cell imaging of RPE1 or HeLaK CHMP4B-LAP-eGFP mCherry-NLS SNAP-LRRC59 cells after MN induction (AZ3146). LRRC59 mean intensity (fold increase) 60 min post-rupture. Error bars: mean ± SEM, 3 independent experiments, dots represent intensity per MN; $n = 36$ (RPE1); 52 cells (HeLaK). **b** LRRC59 accumulates at MN upon rupture. Stills from live-cell imaging of RPE1 CHMP4B-LAP-mNG mCherry-NLS SNAP-LRRC59 cells during MN rupture. SNAP-LRRC59 staining using SiR-SNAP. Arrows, MN. Scale bars, 10 µm; $N = 3$. **c** Quantification of SNAP-LRRC59 intensities at various timepoints following MN rupture (t = 0). Error bars: mean (lines) ± SD (bands), 3 independent experiments: $n = 39$ cells. Arb. units, arbitrary units. **d** The LRRC59 ER luminal domain regulates recruitment to ruptured MN. Quantification of LRRC59-positive ruptured MN fraction in RPE1 cells expressing siRNA #3-resistant SNAP-LRRC59. Error bars: mean ± SEM, 3 independent experiments; $n = 53$ (LRRC59$^{FL}$); 55 (LRRC59$^{\Delta LUM}$); 44 cells (LRRC59$^{\Delta NLS}$). LRRC59$^{FL}$ vs LRRC59$^{\Delta LUM}$, *$P = 0.0160$; LRRC59$^{FL}$ vs LRRC59$^{\Delta NLS}$, NS, $P = 0.4671$. One-way ANOVA of AUC with Dunnett's test. **e** Endogenous LRRC59 and CHMP7 associate at ruptured MN. Representative confocal images of RPE1 CHMP4B-LAP-mNG mCherry-NLS cells treated with AZ3146. PLA antibodies against CHMP7 and LRRC59; DNA counterstain (Hoechst). Scale bars, 10 µm; $N = 3$. **f** LRRC59 depletion triggers elevated and accelerated accumulation of LEMD2 to ruptured MN. Stills from live-cell imaging of LEMD2-APEX2-mCitrine RPE1 cells treated with AZ3146 and indicated siRNAs. Arrows indicate MN. Scale bars, 10 µm; $N = 3$. **g** Quantification of LEMD2 fold increase at ruptured MN. Error bars: mean

(lines) ± 95% CI (bands), 3 independent experiments; $n = 28$ (siCon); 15 (siLRRC59 #1); 49 cells (siLRRC59 #2). siCon vs siLRRC59 #1, ****$P < 0.0001$; siCon vs siLRRC59 #2, ****$P < 0.0001$. One-way ANOVA of AUC with Dunnett's test. **h** LRRC59 depletion increases MN repair frequency in RPE1 cells. Quantification of repaired MN fraction from live-cell imaging of RPE1 CHMP4B-LAP-mNG mCherry-NLS cells treated with indicated siRNAs. Error bars: mean ± SEM, 4 independent experiments, dots represent mean/experiment; $n = 125$ (siCon); 107 (siLRRC59 #1); 77 cells (siLRRC59 #3). siCon vs siLRRC59 #1, ****$P < 0.0001$; siCon vs siLRRC59 #3, ****$P < 0.0001$. One-way ANOVA with Dunnett's test. **i** Overexpression of LRRC59 decreases MN repair frequency in RPE1 cells. Quantification of repaired MN fraction from live-cell imaging of RPE1 CHMP4B-LAP-mNG mCherry-NLS cells with DOX-inducible SNAP-LRRC59, treated with DOX. MN rupture and repair events were assessed at indicated hours after SNAP-LRRC59 induction. Error bars: mean ± SEM, 3 independent experiments, dots represent mean/experiment; $n = 65$ (0 h); 85 (24 h); 125 cells (48 h). 0 h vs 24 h, **$P = 0.0014$; 0 h vs 48 h, **$P = 0.0063$. One-way ANOVA with Dunnett's test. **j** LRRC59 depletion increases DNA damage in ruptured MN. Quantification of γH2Ax-positive MN fraction from RPE1 mCherry-NLS cells treated with indicated siRNAs. Error bars: mean ± SEM, 3 independent experiments, dots represent mean/experiment; $n = 93$ (siCon intact); 106 (siLRRC59 intact); 98 (siCon ruptured); 133 cells (siLRRC59 ruptured). siCon intact vs siLRRC59 #1 intact, NS, $P = >0.9999$. siCon ruptured vs siLRRC59 ruptured, **$P = 0.0091$. One-way ANOVA with Bonferroni's test. Source data are provided as a Source Data file.

overexpression might introduce biases in our proteomics setup, the fact that we identify numerous (14 proteins) ESCRT-III proteins and associated factors known to regulate NE sealing[22,41,48] provides a key validation of our approach. Among these, CC2D1A and the protease Calpain 7 (CAPN7) have not previously been implicated at the NE (Fig. 2e), indicating the importance of further ESCRT-associated

modalities during NE repair[49]. The presence of a large ubiquitylation node centered on the USP8[50] and STAMBP/AMSH[24] deubiquitylases (Fig. 2e) could reflect regulation of ESCRT-III kinetics at NE lesions[72]. Alternatively, it could contribute to the interplay between ESCRT-III and the autophagic machinery at ruptured MN[38,39] that exist at a low baseline frequency in RPE1 cells.

In addition to ESCRT-associated nodes, we identify numerous proteins that constitute a large ER morphogenesis cluster that can broadly be differentiated into ER-shaping[51,52,73] and phospholipid biogenesis nodes (Fig. 2e). De novo membrane biogenesis, composition and curvature all directly regulate ESCRT function at the NE[74–77]. Surprisingly, we did not find the involved membrane biogenesis regulators Lipin1 and CTDNEP1[45] in our proteomics. Instead, we identify the acyl-CoA synthetase 3 (ACSL3) and the lysophosphatidylcholine acyltransferase 1 (LPCAT1), two enzymes critical to the synthesis of phosphatidylcholine from lysophosphatidylcholine (LPC)[75,78,79]. This distinction could reflect the limited need for de novo membrane biogenesis during interphase NE repair versus the extensive nuclear membrane expansion involved in mitotic NE reformation.

The LINC component SYNE2 was recently shown to bind the ESCRT-III associated protein BROX at the NE, suggesting it could contribute to ESCRT-III dependent NE repair[41]. Interestingly, our NE repairome identifies a large actin network centered around the SYNE2 homolog SYNE1, in addition to BROX. We further identify the formin-like FHOD1 (Fig. 2E) that directly associates with SYNE1 to bundle and stabilize perinuclear actin filaments and thereby control nuclear force transduction[80]. Our data support a model[41] where a stable LINC-FHOD1 network, possibly in cooperation with phospholipid production alleviates membrane tension at the site of rupture, which is a prerequisite for ESCRT-III mediated membrane scission[81,82].

Recent work has indicated that ruptured endolysosomes are plugged with a biomolecular condensate of stress granule proteins and associated RNAs, followed by repair of the damaged membrane through ESCRT-III activity[83]. Our identification of a large cluster of RNA-binding and stress granule proteins[84] (Fig. 2e) therefore raises the possibility that these factors form an analogous biomolecular condensate at NE ruptures. This would allow restoration of compartmentalization and possibly even control membrane influx[85] prior to final sealing of the membrane annulus by ESCRT-III. It would also be consistent with our observations that CHMP2A depletion abrogates ESCRT-III dynamics at NE ruptures with minimal effects on nucleoplasmic mCherry-NLS restoration (Fig. 5), further arguing that nuclear compartmentalization and NE resealing are separate events. Further precedent for such a mechanism comes from recent work in fission yeast showing that perturbation of ESCRT-III function does not block nuclear recompartmentalization despite the presence of large membrane discontinuities (500 nm diameter) around the spindle body[60].

Within our NE repairome, we identify LRRC59 as a prominent regulator of the LEMD2-CHMP7 compartmentalization sensor. We show that LRRC59 together with KPNB1 controls accumulation and turnover of LEMD2-CHMP7 at NE lesions in synergy with XPO1, adding an additional layer of regulation of NE rupture-repair cycles (Figs. 3, 5, and 6). Based on our data, we propose the following model of NE repair (Supplementary Fig. 9). Under normal conditions, CHMP7 localization is restricted to the ER by XPO1-mediated nuclear export, spatially separating it from the INM protein LEMD2[18]. ER-integral LRRC59 is imported to the INM by KPNB1, where it dissociates from KPNB1 and subsequently binds to LEMD2 through its ER luminal domain[54] (Fig. 3). NE rupture locally dissipates the RAN gradient[65], causing CHMP7 dissociation from XPO1, and allowing CHMP7 to bind to LEMD2[18,23]. At the same time, the compromised RAN gradient prevents dissociation of KPNB1 from LRRC59 at the INM. As the NLS is juxtaposed to the transmembrane and luminal domains of LRRC59 (Fig. 6h), we propose that the continued association with KPNB1 sterically interferes with the binding of LRRC59 to LEMD2 (Supplementary Fig. 7). These two events regionally license the formation of a LEMD2-CHMP7 biomolecular condensate[23] that drives nucleation of ESCRT-III filaments and ultimately allows the resealing of the NE. Restoration of the RAN gradient following NE resealing and completion of ESCRT-III function triggers reassociation of XPO1 with CHMP7, dissociating it from LEMD2 and translocating it to the ER. Concurrent dissociation of LRRC59 from KPNB1 allows it to bind

LEMD2 and further drive active disassembly of the LEMD2-CHMP7 biomolecular condensate, thereby restoring the pre-rupture state (Supplementary Fig. 9). Simultaneous perturbation of LRRC59-KPNB1 and XPO1 inactivate the dual restraints to LEMD2 accumulation into a biomolecular condensate, causing it to spread from the NE lesion along an extensive part of the INM (Fig. 7a and c, Supplementary Fig. 6E and 6F). Spread of this LEMD2 condensate[23] causes extensive DNA torsional stress in its wake (Fig. 7f). Consistent with LEMD2-CHMP7 assembly in this process, CHMP4B accumulation in foci along the leading edge of this wave indicates the activation of downstream ESCRT-III polymerization. The fact that CHMP4B foci appear only transiently along the leading edge of the spreading LEMD2-CHMP7 supports our data that LRRC59 regulates the LEMD2-CHMP7 interaction without affecting downstream ESCRT-III dynamics (Figs. 5 and 6). Previous work has shown regulation of CHMP7-LEMD2 interaction during mitosis by cyclin-dependent kinase 1 (CDK1)-mediated phosphorylation of CHMP7[16]. Although we do not identify any CDKs in the NE repairome, it will be very interesting to see whether the LRRC59-KPNB1 and XPO1 nodes intersect with CDK-dependent mechanisms during NE reformation and interphase NE repair (Figs. 4–6).

Besides its role in PN integrity, LRRC59 has dual effects on rupturing MN. Firstly, it regulates the probability for successful repair of rupturing MN (Fig. 8h), by shifting the balance between LEMD2 and CHMP7, similar to what we previously observed for partial CHMP7 depletion[18]. Secondly, in the subpopulation of non-repairing MNs, LRRC59 depletion caused an accelerated enrichment and spread of LEMD2 across the MN. Importantly, we find that this is accompanied by increased levels of DNA damage (Fig. 8j), drawing parallels to our previous reports of MN catastrophe following unrestrained ESCRT-III activity[18] and our observations of DNA torsional stress in PN[18] (Fig. 7f) upon coordinated inactivation with XPO1. Our ability to recapitulate MN rupture phenotypes in PN through coordinated perturbation of LRRC59-KPNB1 and XPO1-CHMP7 is striking and provides a compelling mechanistic model explaining the repair defects and DNA damage accumulation that dominate ruptured MN.

It is well-established that MN frequently display impaired nuclear import and export, as well as altered protein composition[18,67,68,86]. As a result, it is likely that the phenotypic dichotomy of LRRC59 depletion reflects high heterogeneity in MN protein composition[67] and varying competence for functional XPO1- and KPN-dependent functions[18,67,68]. Alternatively, this dichotomy could reflect differential levels of ROS and recruitment of the autophagy machinery that control CHMP7 activity at ruptured MN[38,39]. While we did identify a few autophagy-related factors in the NE repairome of PN, the key autophagy marker p62 was not amongst these (Fig. 2). Nevertheless, we find very similar GO terms as those recently observed in proteomics of ruptured MN[39]. Intersecting these datasets is likely to uncover further functional parallels and distinctions between PN and MN rupture models and their disparate repair propensity.

High LRRC59 expression has been shown to associate with poor prognosis in several cancer types[87,88]. Our implication of LRRC59 in the repair of ruptured PN and MN provides a plausible rationale for these observations that warrants further exploration. Furthermore, many tumors are characterized by altered (micro-)nuclear import and export[19,65,89], and our data suggest that NE repair capacity is likely to be perturbed under these conditions. Finally, the use of export (XPO1) and import (KPNB1) inhibitors is explored in chemotherapeutic treatment of several cancer types[19,65,89]. These points may have significant context-dependent positive or negative effects on cancer progression by modulation of PN and MN repair capacity, the level of DNA torsional stress and genome instability, and resulting inflammatory signaling[1,3,8,69]. In conclusion, LRRC59 and other hits from our proteome provide key insights into NE rupture-repair processes, but also inroads to tailored treatment of cancer and other diseases characterized by compromised nuclear stability.

## Methods

### Cell culture

The cell lines used in this study were cultured according to the American Type Culture Collection (ATCC). RPE1-hTERT cells (ATCC CRL-4000) were cultured in DMEM/F-12 with GlutaMAX™ (Sigma-Aldrich). Hela K and HEK293T cells were cultured in DMEM with high glucose (4500 mg l$^{-1}$), GlutaMAX™, and sodium pyruvate (110 mg l$^{-1}$) (Sigma-Aldrich). All media were supplemented with 10% Fetal bovine serum (FBS) (Gibco) and 1% Penicillin-Streptomycin (PenStrep; Sigma-Aldrich). The cells were maintained at 37 °C with 5% CO$_2$. The stable cell lines expressing inducible CHMP7 (wild type (WT), NLS, or NES*) or LRRC59 alleles were treated with 500 ng ml$^{-1}$ of doxycycline (DOX; Sigma-Aldrich) to induce the expression of the desired transgenes. Micronucleation was induced by treating the cells with 2-4 μM AZ3146 (Selleckchem). To arrest the cells in mitosis, cells were treated with 5 μM S-Trityl-L-cysteine STLC (Sigma-Aldrich). To inhibit Exportin 1 (XPO1) function, cells were incubated with 5 ng ml$^{-1}$ Leptomycin B (Sigma-Aldrich). To block nuclear import via importin-β, cells were treated with 50 μM Importazole (Merck).

### Plasmids and generation of stable cell lines

Stable cell lines were generated in this study using lentivirus-generated pools[18,90]. The expression level of transgenes was controlled by using different promoters - the weak PGK promoter for low expression and the CMV or EF1α promoters for high expression. Lentiviral transfer vectors were constructed by generating Gateway ENTRY plasmids of CHMP4B-APEX2-mCitrine, LEMD2-APEX2-mCitrine, mCherry-NLS, LRRC59-V5-TurboID, and SNAP-LRRC59 alleles. These plasmids were then recombined into lentiviral destination vectors, which were Gateway-enabled vectors derived from pCDH-EF1a-MCS-IRES-PURO. The lentiviral particles were produced using a third-generation packaging system, with plasmids (12251, 12253, and 12259) obtained from Addgene. To prevent overexpression, cells were transduced with a low virus titer (multiplicity of infection <1). The medium was replaced the next day, and antibiotic selection was initiated to establish stable cell populations that express the desired transgenes. Supplementary Table 1 provides a list of the specific stable cell lines used in this study. Information regarding constructs used in this study is available upon request.

### Antibodies

Primary antibodies used were mouse anti-CHMP7 (Abnova H00091782-B01P, lot m5041; IF, 1:100; PLA, 1:200), rabbit anti-CHMP7 (Proteintech 16424-1-AP, lot 00039615; WB, 1:1000), mouse anti-FLAG (Sigma-Aldrich, F1804, lot SLBS3530V; PLA, 1:750; WB, 1:2000), rabbit anti-LRRC59 (Sigma-Aldrich, HPA030829, lot A105279; IF, 1:300; PLA, 1:250; WB, 1:1000), mouse anti-β-actin (Proteintech, 66009-1-Ig, lot 10025459; WB, 1:10000), rabbit anti-LaminB1 (Abcam, ab16048, lot GR3459550-1; WB, 1:1000), rabbit anti-LEMD2 (Sigma-Aldrich, HPA017340, lot 000007624; WB, 1:1000; IF, 1:200), mouse anti-Top2b (SCBT, sc-25330, lot I1817; IF, 1:100), rabbit anti-GFP (Abcam, ab32146, lot GR253725-25; WB, 1:5000), mouse anti-eGFP/mCitrine (Roche, 11814460001, lot 19958500; IF, 1:750; PLA, 1:300; WB, 1:1000), mouse anti-γH2Ax (Millipore, 05-636, lot 3761799; IF, 1:500), rabbit anti-SNAPtag (ThermoFisher CAB4255, lot XC343633; WB, 1:1000), mouse anti-LBR (Abcam ab232731, lot GR3353170-6; WB, 1:1000), mouse anti-Lamin A/C (SCBT sc-7292, lot H0720; WB, 1:500), mouse anti-p21/CDKN1A (SCBT sc-6246, lot B2820; IF, 1:200), mouse anti-Ki67 (SCBT sc-23900, lot F0118; IF, 1:200), Streptavidin-AF647 (ThermoFisher, s21374, lot 1924460; IF, 1:500), Streptavidin-HRP (ThermoFisher, s911, lot 1880067; WB, 1:5000). Secondary antibodies included anti-mouse- and anti-rabbit -Alexa488, -Alexa568, and -Alexa647 (all Molecular Probes), -IRDye680, -IRDye800 (LI-COR), and -HRP (Jackson) conjugates.

### APEX2 proximity-labeling, purification, and LC-MS/MS analysis

For protein identification by mass spectrometry, cells expressing APEX2-fusions were incubated with complete medium supplemented with 500 μM biotin-phenol (Iris-Biotech) at 37 °C for 3 h. The medium was then replaced with 2 mM H$_2$O$_2$ in PBS and incubated for 2 minutes at room temperature. To quench the reaction, a solution containing 5 mM Trolox (Sigma-Aldrich) and 10 mM sodium ascorbate (Sigma-Aldrich) was added. This quencher solution was used for three additional washes. The cells were then lysed on the dish using RIPA buffer (50 mM Tris-HCl pH 7.5, 150 mM NaCl, 1% Triton X-100, 0.1% SDS, 0.5% NaDOC, 5 mM EDTA, 1 mM DTT, and protease inhibitors) and placed on ice for 15 min. The samples were centrifuged at 4 °C for 15 minutes, and the supernatant was transferred to another tube. To remove excess non-conjugated biotin-phenol, cell lysates were desalted using Zeba spin columns with size exclusion limit of 7 kDa, according to the manufacturer's protocol (ThermoFisher Scientific). Following desalting, the samples were incubated with streptavidin magnetic beads (Invitrogen Dynabeads M280) at 4 °C using end-over-end rotation for 2–3 h. After magnetic isolation, bound material was washed twice with RIPA buffer, twice with 0.1% Triton X-100 in PBS, three times with 1% SDS in PBS, twice with 4 M Urea in PBS, and five times with PBS. The samples were then delivered for quantitative tandem mass spectrometry processing in 200 μl of PBS or processed for western blot analysis by elution in 2x Laemmli buffer (125 mM Tris-HCl pH 6.8, 4% SDS, 20% glycerol, 20 mM DTT, and bromophenol blue) containing 2 mM biotin (Sigma-Aldrich).

Quantitative tandem mass spectrometry and analysis was done as described (Perseus (1.6.15)) and was visualized using Cytoscape (3.10.2)[91]. To compile a shortlist of high-probability significant candidate hits, we required proteins to meet stringent enrichment thresholds for both LEMD2-APEX2 and CHMP4B-APEX2 preps, comparing before and after CHMP7 allele induction. These enrichment thresholds were based on observed CHMP4B enrichment in LEMD2-APEX2 preps and vice versa. For CHMP7$^{WT}$ inductions, these thresholds were 4-fold enrichment (in LEMD2-APEX2 preps) and 6-fold (CHMP4B-APEX2). For CHMP7$^{NLS}$, thresholds were 3-fold enrichment (in LEMD2-APEX2 preps) and 4.5-fold (CHMP4B-APEX2). For the CHMP7$^{NES*}$ induction time course, the thresholds were 2.5-fold enrichment (in LEMD2-APEX2 preps) and 8-fold (CHMP4B-APEX2) for at least 1 time point. This resulted in a shortlist of 110 proteins enriched upon induction of all 3 CHMP7 alleles (Supplementary Table 1). In addition, we determined chromatin-dependent interactors by comparing CHMP7$^{NLS}$ vs CHMP7$^{WT}$ (5.5-fold relative enrichment), and we identified an additional 2 factors, namely Top2B and VRK1. GO analyses[92,93] (Fig. 2C) reflect Bonferroni-corrected p-values (-log10), with color scaling intensities reflect protein count.

### Immunostaining

The cells were cultured on 12 mm round coverslips (VWR, cat. no. 631-1577) and fixed with a 4% formaldehyde solution in PEM buffer (80 mM Pipes, 5 mM EGTA, 1 mM MgCl2, pH 6.8) for 10 minutes at room temperature. Following fixation, the cells were permeabilised with 0.2% Triton X-100 in PEM buffer for 2–5 minutes at room temperature. Primary and secondary antibodies were diluted in PBS containing 0.01% Tween 20 (PBST) and incubated with the cells for 1 hour to overnight. Several washes with PBST were performed in between each antibody staining step. After antibody staining, the samples were mounted onto microscope slides (Epredia) in Mowiol (Sigma) containing Hoechst 33342 (1 μg ml$^{-1}$, Sigma-Aldrich) as a nuclear counterstain.

### Bi-molecular fluorescence (BiFC) assay

Coding sequences of CHMP7 and LRRC59 were cloned into plasmids containing either the N-terminal (VN) or C-terminal (VC) fragment of Venus fluorescent protein to produce pCHMP7-mVC155, pVN155(I152L)-

LRRC59, and pmVC155 (for control). HeLa K cells were seeded onto 12 mm coverslips (VWR) placed in 6-well plates. Transient transfection was performed with Fugene 6 (Promega) according to the manufacturer's protocol, with co-transfection of an mRuby3-NES plasmid as a transfection control. After 20 hours, cells were fixed and processed for immunostaining as indicated above.

## Proximity ligation assay

The cells were cultured on 12 mm coverslips (VWR, cat. no. 631-1577) and fixed and processed as above. Proximity ligation assays (PLA) were conducted using the Duolink® PLA reagents (Sigma-Aldrich), according to manufacturer's protocol. The coverslips were then blocked with Duolink® Blocking Solution for 60 min at 37 °C. Following blocking, cells were incubated overnight at 4 °C with primary antibodies in Duolink Antibody Diluent (1X). After incubation, the cells were washed three times with 1x Buffer A. The cells were then incubated with PLA probe solution (Duolink Antibody Diluent (1X), PLA-anti-Rabbit PLUS, and PLA-anti-Mouse MINUS, both at a dilution of 1:5) for 1 hour at 37 °C. Following probe incubation, cells were washed three times with 1x Buffer A. For the ligation step, cells were incubated with Ligation-Ligase solution (a ligase ratio of 1:40 in ligation buffer) for 30 minutes at 37 °C and washed three times with 1x Buffer A again. For signal amplification, cells were treated with Amplification (Detection Reagents FarRed)-Polymerase solution (a polymerase ratio of 1:80 in amplification buffer) for 100 minutes at 37 °C. Post amplification, cells were washed three times with 2x Buffer B and once with 0.01x Buffer B. Finally, coverslips were mounted onto microscope slides (Menzel-Glaser) using Mowiol (Sigma-Aldrich) containing Hoechst 33342 (1 µg ml⁻¹, Sigma-Aldrich) as a nuclear counterstain.

## Confocal (fluorescence) microscopy

Fixed cell samples were imaged using a Dragonfly 505 spinning disk confocal microscope (Andor Technology), equipped with a Zyla 4.2 PLUS sCMOS camera. Imaging was conducted using either a 60×/1.40NA oil immersion objective or a 100×/1.49NA oil immersion objective. Images were processed and analyzed using ImageJ software.

## Live-cell imaging

For live-cell imaging, cells were seeded into Lab-Tek chambered coverslips (Nunc). Environmental control during live observation was maintained using a temperature-controlled incubation chamber. Cells were imaged in DMEM medium without phenol red (Gibco), supplemented with 10% fetal bovine serum, 1% penicillin–streptomycin, 1x GlutaMax (Gibco), and 25 mM HEPES (Sigma-Aldrich). SiR-Hoechst (50-100 nM; SpiroChrome) or SPY650-DNA (5000x; SpiroChrome) was used to visualize DNA, and SNAP-Cell 647-SiR (SiR-SNAP, 0.5 µM; New England Biolabs) was employed to detect SNAP-tagged alleles. Imaging was conducted using a Dragonfly 505 spinning disk confocal microscope (Andor Technology), equipped with an iXon Life 888 EMCCD camera. A 60×/1.40NA oil immersion objective was used for high-resolution image acquisition. Cells were imaged every 2 min for 6 h to detect LEMD2 at the reforming NE after mitosis, every 2 min for 12 h to monitor NE ruptures and ESCRT recruitment, every 10 min for 12 h to quantify LEMD2 foci size following CHMP7 allele overexpression using DOX, every 5 minutes for 12 hours to quantify LEMD2 accumulation at ruptured micronuclei, and every 2.5 or 5 minutes for 12 or 16 hours (HeLaK or RPE1 cells respectively) to quantify LRRC59 accumulation at ruptured micronuclei.

## siRNA transfections

All siRNAs used in this study were purchased from ThermoFisher and contained the Silencer Select modification. Cells were transfected using Lipofectamine RNAiMAX transfection reagent (ThermoFisher) according to the manufacturer's instructions. A total siRNA concentration of 40 nM was used for each transfection. Cells transfected with siRNAs targeting *LRRC59* (#1 AAGGUGUUACAGCACAUGAtt; #2 GATCAGGAGCGGGAGAGGCAtt; #3 GAGUAUGAUGCCCUCAAAGtt), *CHMP7* (AGGUCUCUCCAGUCAAUGAtt), *LMNB1* (cat. no. s8225), or *KPNB1* (cat. no. s7919) and were imaged 48-72 hours post-transfection. Cells transfected with siRNA targeting *CHMP2A* (AAGAUGAAGAGGA-GAGUGAtt) were imaged 36 hours post-transfection. Non-targeting control Silencer Select siRNA (cat. no. 4390844) was used as a control.

## Co-immunoprecipitation

HEK293T cells cultured in 10 cm dishes were cotransfected with plasmids expressing KNPB1-eGFP (Addgene #106941) and SNAP-LRRC59^ΔC or SNAP-LRRC59^{ΔC,ΔNLS} for 48 h. During the final 7 h, 50 µM Importazole was added to the medium to stabilize KPNB1 association with cargos. Cells were washed with PBS, extracted in lysis buffer (50 mM HEPES pH 7.5, 100 mM KCl, 0.1% Triton X100, 1 mM MgCl₂, 1 mM EGTA, protease inhibitors) and centrifuged for 20 min at 20,000 × g and 4 °C. Cleared lysates were incubated with 10 µl washed magnetic GFP-trap beads (Proteintech, gtd-20) and immunoprecipitations were performed at 4 °C for 2–3 h with end-over-end rotation. Beads were washed 4 times in lysis buffer after which 2x Laemmli buffer (125 mM Tris-HCl pH 6.8, 4% SDS, 20% glycerol, 20 mM DTT, and bromophenol blue) was added to the beads and boiled for 20 minutes. Samples were analysed by immunoblotting.

## Immunoblotting

The cells were lysed using 2x Laemmli buffer (125 mM Tris-HCl pH 6.8, 4% SDS, 20% glycerol, 20 mM DTT, and bromophenol blue). The whole-cell lysates were then boiled at 95 °C for 10 minutes and separated by SDS-PAGE using a 4-15% gradient gel. Following gel electrophoresis, the proteins were transferred to a FL PVDF Membrane (Bio-Rad, 1704274). The membrane was then blocked with a 5% milk-PBST solution and subsequently incubated with a primary antibody at 4 °C overnight. Proteins were detected using either IRDye- (IRDye680 and IRDye800; LI-COR) or HRP-coupled secondary antibodies in combination with an ECL kit (Thermo Scientific). The ChemiDoc developer (Bio-Rad) was used for visualization. Uncropped western blots are shown in Supplementary Fig. 10 and Source Data File.

## TurboID labeling

To identify interaction partners of LRRC59 within the ER lumen, cells expressing LRRC59-TurboID fusion proteins were incubated with 100 µM biotin (Sigma) at 37 °C for 30 minutes. Subsequently, the cells were washed three times with ice-cold PBS. Cells were lysed directly on the culture dish using RIPA buffer (50 mM Tris-HCl (pH 7.5), 150 mM NaCl, 1% Triton X-100, 0.1% SDS, 0.5% NaDOC, 5 mM EDTA, 1 mM DTT, and protease inhibitors). Lysates were centrifuged at 4 °C for 15 minutes at 13,000 g to separate the supernatant. To remove excess biotin, cleared lysates were desalted using Zeba spin columns with a 7 kDa size exclusion limit according to the manufacturer's instructions (Thermo Fisher Scientific). After desalting, samples were incubated with streptavidin magnetic beads (Invitrogen Dynabeads M280) at 4 °C with end-over-end rotation for 2-3 hours to capture biotinylated proteins. After magnetic isolation, the beads were washed four times with RIPA buffer to ensure removal of non-specific binders. For elution, 2x Laemmli buffer (125 mM Tris-HCl pH 6.8, 4% SDS, 20% glycerol, 20 mM DTT, and bromophenol blue) containing 4 mM biotin (Sigma) was added to the beads and boiled for 20 minutes. Samples were analysed by immunoblotting.

## RT-qPCR

RNA was extracted from cells utilizing the RNeasy Mini Kit (Qiagen) following the manufacturer's protocol. Complementary DNA (cDNA) was synthetized using the High-Capacity cDNA Reverse Transcription Kit (ThermoFisher). Real-time quantitative PCR (qPCR) analysis was conducted with iQ SYBR® Green Supermix (Bio-Rad) on a CFX-96 thermocycler (Bio-Rad, Version 5.0.021.0616). The fold change in gene

expression was calculated using the 2^-ΔΔCT method and normalized to the expression level of *SF3A1*. Oligonucleotide primers (Integrated DNA Technologies, custom design, desalted) used were: *SF3A1*, AGGGTCCAGTGTCCATCAAA (Fw), AGAGACCTGGTCCGTGAGTG (Rv); *CCNB1*, CATGGTGCACTTTCCTCCTT (Fw), AGGTAATGTTGTA-GAGTTGGTGTCC (Rv); *MKI67*, GAAAGAGTGGCAACCTGCCTTC (Fw), GCACCAAGTTTTACTACATCTGCC (Rv).

## Image processing

For the Bi-molecular Fluorescence Complementation (BiFC) assay, cells were segmented in FIJI 1.54f[94] using Otsu thresholding based on the mRuby3-NES channel, and the mean intensity in the BiFC channel was measured. For the proximity ligation assay (PLA), cells were similarly segmented, and PLA puncta within each ROI were identified and quantified using the 'Count Maxima' function, with tolerance values set for each experiment. For the quantification of intranuclear LEMD2 tubules, nuclei were segmented using Otsu thresholding based on the DNA channel, and tubules within these ROIs were identified using predefined intensity and size thresholds. To assess LEMD2 recruitment to the reforming NE during mitosis, cells were arrested with STLC, imaged from wash-out, and nuclei were segmented using Otsu thresholding based on the DNA channel. The mean intensity of the LEMD2 channel in nuclear ROIs was measured, with a band-shaped ROI around the nucleus as a regularization factor for cytoplasmic background intensity. NE ruptures were detected by abrupt drops in nuclear mCherry-NLS signal and manually counted to determine rupture fraction, while nuclear herniations were quantified by counting cells with mCherry-NLS and DNA signal-based blebs. For quantifying the recruitment of LEMD2, CHMP4B, and CHMP7 to NE rupture sites, rupture events were manually identified using the nuclear mCherry-NLS or cytoplasmic mRuby3-NES marker. Fixed-diameter ROIs were drawn at rupture sites and tracked over time, with pre-rupture ROIs used for signal normalization and cytoplasmic ROIs as regularization factors. For quantifying LEMD2 foci size after CHMP7 induction, nuclei were segmented using Otsu thresholding based on the DNA channel, and LEMD2 foci within these ROIs were identified using predefined intensity and size thresholds. The number of LEMD2 foci per cell was quantified and normalized by the mean number of foci from one replicate to account for experimental variations. To quantify rupture repair time, a modified version of the pipeline described by Robijns et al.[59]. was employed. Briefly, a FIJI script (TrackRuptures.ijma) was used to segment and track nuclei over time. Image preprocessing included intensity normalization of sum projections over time to account for temporal variations. Nuclei were detected using StarDist[95] based on the DNA channel. Individual nuclei were connected through time using a nearest neighbor algorithm, confined by maximum displacement and tolerant to gaps of limited time span. After manually selecting tracks where ruptures occurred, signal intensities were extracted and analysed using RStudio (4.1.2 and 4.4.1). The ratio of the mCherry-NLS signal to the DNA signal was calculated, and rupture events were detected when the derivative of this ratio exceeded 1.10. Individual rupture tracks were extracted, normalized, and synchronized to the pre-rupture timepoint. An exponential curve was then fitted to each rupture event to calculate the repair half-time. To quantify the spread of LEMD2 over ruptured nuclei we used CellProfiler 4.2.6[96]. The DNA channel was used to perform nuclear segmentation (due to high variation in DNA signal we used adaptive thresholding using the three-classes Otsu method with a correction factor of 0.8, and with pixels presenting middle intensities being considered as foreground), nuclei touching the image border were not considered. Segmented nuclei were then tracked (using the overlap method and 50 pixels as maximal distance). To avoid signal coming from the nuclear envelope we used eroded nuclei (segmented nuclei eroded by 4 pixels) to mask LEMD2-mCitrine channel. Only nuclei with LEMD2 signal were further analysed. LEMD2 spread signal was segmented by combining

segmentation of small spread instances (size between 2 and 25 pixels and threshold above 0.7) with bigger ones (size between 26 and 600 pixels, thresholded using the three-classes Otsu method with a correction factor of 0.8 and pixels presenting middle intensities being considered as foreground). The extent of LEMD2 spread was calculated nuclear occupancy [(LEMD2 spread area/ Nucleus area) x 100]. Segmentation images were then overlayed with raw images to control the quality of both segmentation and cell tracking. Some tracks that had been wrongly identified as separate (but after visual inspection was clear that corresponded to the same nucleus) were unified. Individual rupture tracks were extracted and synchronized to the pre-rupture timepoint, and results were plotted as percentage of LEMD spread over time after applying a rolling average (with a window width3 of 4). For every track, we calculated the first derivative of the percentage of LEMD spread over time and then selected the maximum first derivative per track for further calculating spreading speeds. R 4.4.1 and RStudio 2024.04.2 + 764 were used for data curation and visualization. The measurement of maximal nuclear occupancy of LEMD2 enrichment following NE rupture was performed using ImageJ (2.16.1/1.54p, Java 1.8.0_322) through a custom script specifically designed to automate processing steps and enable standardized measurements. Time-lapse images were opened and analysed to identify regions of maximal LEMD2 nuclear occupancy. The script prompted the user to define a region of interest (ROI) around the ruptured nuclei at the corresponding time frame; this ROI was subsequently duplicated and saved for documentation purposes, together with its centroid xy coordinates and the corresponding frame and file name. To isolate structures of interest, the images were then thresholded using the RenyiEntropy autothreshold option for the nuclear channel and the MaxEntropy option for the LEMD2 channel. The user had the opportunity to visually adjust the thresholds if necessary before applying them, ensuring accurate segmentation of the targeted structures. Particle analysis was then performed on the segmented images, with size filters implemented to exclude debris and artifacts. For the nuclear channel, analysis was restricted to particles larger than 60 microns, while particles in the LEMD2 channel were analysed when larger than 0.5 microns. Resulting particles from both the nuclear and LEMD2 channels were recorded and saved in the ROI manager. Area measurements were compiled into CSV files for statistical analysis, adhering to a naming convention that ensured clarity and prevented overwriting of existing files. Detected ROIs were archived as ZIP files for convenient retrieval and further evaluation. This process was repeated for each image until all ruptured nuclei had been processed. Subsequent data exploration and curation involved a thorough review of the CSV tables, saved images, and ROIs to verify that only one rupture was identified per nucleus. In the final curated CSV tables, the nuclear occupancy of LEMD2 was calculated as the ratio of LEMD2 spread area to nuclear area [LEMD2 spread area/nuclear area]. These values were used for graphical representation and statistical analyses of the results.

## Molecular modeling

Modeling of LEMD2 (Q8NC56), LRRC59 (Q96AG4), KPNB1 (Q14974) and KPNA2 (P52292) was performed with AlphaFold3 (https://alphafoldserver.com/) without the input of any coordinate files. The parts of the proteins that are facing the nucleoplasm, the ER-luminal side, and the transmembrane helices, were deduced from public database annotations, literature[23,54], and experiments (Fig. 3) and modeled each separately to avoid modeling of unrealistic contacts between those parts. The final models for full-length proteins were reconstructed by manually assembling all parts for each of the molecules in Coot (1.1.15)[97]. The ER-luminal parts of LEMD2 and LRRC59 were modeled as a complex with ipTM=0.52 and pTM=0.67. Each model was minimized using the model minimization software in Phenix (1.21.2)[98]. The figures were created with ChimeraX (1.8)[99]. Modeling output indicates that the relative orientation of KPNA2-KPNB1 is

flexible, and the presented model (Supplementary Fig. 7) represents only one of the possibilities. Nevertheless, our modeling suggests that a distance of 40 Å between the TM helices (persist as helices into the nucleoplasm, projecting 30-40 Å beyond the membrane) of LEMD2, which persist as helices into the nucleoplasm, is sterically incompatible with the binding of the LRRC59-KPNA2-KPNB1 complex, regardless of the chosen KPNA2-KPNB1 orientation (with a diameter >50 Å right below the membrane).

## Statistical analysis

All experiments are representative of at least three independent experimental replicates. Details regarding statistical significance and sample size are included in the respective figure legends. For statistical analysis, p-values were interpreted as follows: NS, $P > 0.05$ represents no significant difference, $* P < 0.05$, $** P < 0.01$, $*** P < 0.001$, and $**** P < 0.0001$. Statistical analysis was performed using Graphpad Prism 10 (GraphPad, CA, USA). Unless otherwise indicated, Student's t-test was used as a measure for statistical significance when comparing two groups, whereas ANOVA was used for comparisons involving multiple groups.

## Reporting summary

Further information on research design is available in the Nature Portfolio Reporting Summary linked to this article.

## Data availability

All data shown and used to generate plots for this manuscript can be found in the source data file. Uncropped western blots are shown in Supplementary Fig. 10. The proteomics data generated in this study have been deposited in public repository ProteomeXchange under PRIDE identifier PXD058192. All other reagents and data will be made available upon request to the lead author. Source data are provided with this paper.

## Code availability

All code is available via Github, https://github.com/DeVosLab.

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

## Acknowledgements
We thank Antoni Wiedlocha and Ellen Haugsten for the LRRC59 constructs and Kay Schink for the critical reading of the manuscript. We are grateful to A. A. Hyman for the gift of HeLa CHMP4B–eGFP BAC cells. We thank the MIP Advanced Light Microscopy Core Facility at the Institute for Basic Medical Science (University of Oslo) for providing imaging facilities. Mass spectrometry-based proteomic analyses were performed by the Proteomics Core Facility, Department of Immunology, University of Oslo/Oslo University Hospital, which is supported by the Core Facilities program of the South-Eastern Norway Regional Health Authority. This core facility is also a member of the National Network of Advanced Proteomics Infrastructure (NAPI), which is funded by the Research Council of Norway INFRASTRUKTUR-program (project number: 295910). This study was supported by a FRIPRO grant of the Research Council of Norway grants to C.C. (grant no. 314655). W.H.D.V. acknowledges support for S.P. from the University of Antwerp (Special Research Fund, N°36910-DOCPRO4_2018; 51172-DOCPRO_2024) and the Flemish Research Fund (FWO, G033322N, I000321N, I003420N).

## Author contributions
Conceptualization, supervision, and funding acquisition: C.C. Experimental design and execution: R.T., A.B., H.K., S.E., A.A., E.O., N.S., C.C. Data analysis: R.T., A.B., S.P., H.K., L.R.B., N.S., W.H.D.V., C.C. Writing: C.C. and R.T., with input from all authors.

## Competing interests
The authors declare no competing interests.
