## [Transparent Peer Review file · Nature Communications]

LRRC59 cooperates with nuclear transporters to restrain the nuclear envelope repair machinery and safeguard genome integrity

Corresponding Author: Professor Coen Campsteijn

This manuscript has been previously reviewed at another journal. This document only contains information relating to versions considered at Nature Communications. Mentions of the other journal have been redacted.

Version 0:

Reviewer comments:

Reviewer #2

(Remarks to the Author)

This is a thoroughly revised manuscript that I had previously reviewed for [REDACTED]. I was somewhat surprised to see it rejected from [REDACTED], considering that the reviewers' only criticisms pertained to some preliminary aspects of the conclusions, rather than the originality or impact of the study.

That said, Nature Communications is a highly respected, high-impact journal, and I am pleased to strongly recommend the current version of this manuscript for immediate publication. The paper has significantly improved during the revision process, notably through the addition of novel data that strengthen the evidence for crosstalk between LRRC59 and KPNB1 and its role in maintaining nuclear envelope integrity.

Reviewer #3

(Remarks to the Author)

The authors have provided substantial new data and analysis to elevate the impact of their conclusions. I'm supportive of publishing this interesting work.
